# Fairness in Ranking under Uncertainty

**Ashudeep Singh**
Cornell University
Ithaca, NY 14850
ashudeep@cs.cornell.edu

**David Kempe**
University of Southern California
Los Angeles, CA 90089
david.m.kempe@gmail.com

**Thorsten Joachims**
Cornell University
Ithaca, NY 14850
tj@cs.cornell.edu

## Abstract

Fairness has emerged as an important consideration in algorithmic decision making. Unfairness occurs when an agent with higher merit obtains a worse outcome than an agent with lower merit. Our central point is that a primary cause of unfairness is uncertainty. A principal or algorithm making decisions never has access to the agents' true merit, and instead uses proxy features that only imperfectly predict merit (e.g., GPA, star ratings, recommendation letters). None of these ever fully capture an agent's merit; yet existing approaches have mostly been defining fairness notions directly based on observed features and outcomes.

Our primary point is that it is more principled to acknowledge and model the uncertainty explicitly. The role of observed features is to give rise to a posterior distribution of the agents' merits. We use this viewpoint to define a notion of approximate fairness in ranking. We call an algorithm $\phi$-fair (for $\phi \in [0, 1]$) if it has the following property for all agents $x$ and all $k$: if agent $x$ is among the top $k$ agents with respect to *merit* with probability at least $\rho$ (according to the posterior merit distribution), then the algorithm places the agent among the top $k$ agents in its *ranking* with probability at least $\phi\rho$.

We show how to compute rankings that optimally trade off approximate fairness against utility to the principal. In addition to the theoretical characterization, we present an empirical analysis of the potential impact of the approach in simulation studies. For real-world validation, we applied the approach in the context of a paper recommendation system that we built and fielded at the KDD 2020 conference.

## 1 Introduction

Fairness is an important consideration in decision-making, in particular when a limited resource must be allocated among multiple agents by a principal (or decision maker). A widely accepted tenet of fairness is that if an agent B does not have stronger merits for the resource than A, then B should not get more of the resource than A. Depending on the context, *merit* could be a qualification (e.g., job performance), a need (e.g., disaster relief), or some other measure of eligibility.

The motivation for our work is that uncertainty about merits is a primary reason that a principal's allocations can violate this tenet and thereby lead to unfair outcomes. Were agents' merits fully observable, it would be both fair and in the principal's best interest to rank agents by their merit. However, actual merits are practically always unobservable. Consider the following standard algorithmic decision making environments: (1) An e-commerce or recommender platform (the principal) displays items (the agents) in response to a user query. An item's merit is the utility the user would derive from it, whereas the platform can only observe numerical ratings, text reviews, the user's past history, and similar features. (2) A job recommendation site or employer (the principal) wants to recommend/hire one or more applicants (the agents). The merit of an applicant is her (future) performance on the job over a period of time, whereas the site or employer can only observe (past) grades, test scores, recommendation letters, performance in interviews, and similar assessments.

35th Conference on Neural Information Processing Systems (NeurIPS 2021).

In both of these examples — and essentially all others in which algorithms are called upon to make allocation decisions between agents — uncertainty about merit is unavoidable, and arises from multiple sources: (1) the training data of a machine learning algorithm is a random sample, (2) the features themselves often come from a random process, and (3) the merit itself may only be revealed in the future after a random process (e.g., whether an item is sold or an employee performs well). Given that decisions will be made in the presence of uncertainty, it is important to define the notion of *fairness* under uncertainty. Extending the aforementioned tenet that "if agent B has less merit than A, then B should not be treated better than A," we state the following generalization to uncertainty about merit, first for just two agents:

**Axiom 1.** If A has merit greater than or equal to B with probability at least $\rho$, then a fair policy should treat A at least as well as B with probability at least $\rho$.

This being an axiom, we cannot offer a mathematical justification. It captures an inherent sense of fairness in the absence of enough information, and it converges to the conventional tenet as uncertainty is reduced. In particular, consider the following situation: two items A, B with 10 reviews each have average star ratings of 3.9 and 3.8, respectively; or two job applicants A, B have GPAs of 3.9 and 3.8. While this constitutes some (weak) evidence that A may have more merit than B, this evidence leaves substantial uncertainty. The posterior merit distributions based on the available information should reflect this uncertainty by having non-trivial variance; our axiom then implies that A and B must be treated similarly to achieve fairness. In particular, it would be highly unfair to *deterministically* rank A ahead of B (or vice versa). Our main point is that this uncertainty, rather than the specific numerical values of 3.9 and 3.8, is the reason why a mechanism should treat A and B similarly.

## 1.1 Our Contributions

We study fairness in the presence of uncertainty specifically for the generalization where the principal must rank $n$ items. Our main contribution is the fairness framework, giving definitions of fairness in ranking in the presence of uncertainty. This framework, including extensions to approximate notions of fairness, is presented and discussed in depth in § 2. We believe that uncertainty of merit is one of the most important sources of unfairness, and modeling it explicitly and axiomatically is key to addressing it.

Next, in § 3, we present algorithms for a principal to achieve (approximately) fair ranking distributions. A simple algorithm the principal may use to achieve approximate fairness is to mix between an optimal (unfair) ranking and (perfectly fair) Thompson sampling. We show that this policy is not optimal for the principal's utility, and we present an efficient LP-based algorithm that achieves an optimal ranking distribution for the principal, subject to an approximate fairness constraint.

We next explore empirically to what extent a focus on fairness towards the agents reduces the principal's utility. We do so with two extensive sets of experiments: one described in § 4 on existing data, and one described in § 5 "in the wild." In the first set of experiments, we consider movie recommendations based on the standard MovieLens dataset and investigate to what extent fairness towards movies would result in lower utility for users of the system. The second experiment was carried out at the 2020 ACM SIGKDD Conference on Knowledge Discovery and Data Mining, where we implemented and fielded a paper recommendation system. Half of the conference attendees using the system received rankings that were modified to ensure greater fairness towards papers, and we report on various metrics that capture the level of engagement of conference participants based on which group they were assigned to.

The upshot of our experiments and theoretical analysis is that in the settings we have studied, high levels of fairness can be achieved at a small loss in utility for the principal and the system's users.

Due to space constraints, essentially all proofs, as well as a detailed discussion of related work, are given in the supplementary material. Alternatively, a reader may want to read the full version (Singh et al., 2021).

## 2 Ranking with Uncertain Merits

We are interested in ranking policies for a principal (the ranking system, such as an e-commerce platform or a job portal in our earlier examples) whose goal is to rank a set $\mathcal{X}$ of $n$ agents (such as

products or applicants). The principal observes some evidence for the merit of the agents, and must produce a distribution over rankings trading off fairness to the agents against the principal's utility. For the agents, a higher rank is always more desirable than a lower rank.

## 2.1 Rankings and Ranking Distributions

We use $\Sigma(\mathcal{X})$ to denote the set of all $n!$ rankings, and $\Pi(\mathcal{X})$ for the set of all distributions over $\Sigma(\mathcal{X})$. We express a ranking $\sigma \in \Sigma(\mathcal{X})$ in terms of the agents assigned to given positions, i.e., $\sigma(k)$ is the agent in position $k$. A ranking distribution $\pi \in \Pi(\mathcal{X})$ can be represented by the $n!$ probabilities $\pi(\sigma)$ of the rankings $\sigma \in \Sigma(\mathcal{X})$. However, all the information relevant for our purposes can be represented more compactly using the *Marginal Rank Distribution*: we write $p_{x,k}^{(\pi)} = \sum_{\sigma:\sigma(k)=x} \pi(\sigma)$ for the probability under $\pi$ that agent $x \in \mathcal{X}$ is in position $k$ in the ranking. We let $\mathcal{P}^{(\pi)} = (p_{x,k}^{(\pi)})_{x,k}$ denote the $n \times n$ matrix of all marginal rank probabilities.

The matrix $\mathcal{P}^{(\pi)}$ is doubly stochastic, i.e., the sum of each row and column is 1. While $\pi$ uniquely defines $\mathcal{P}^{(\pi)}$, the converse mapping may not be unique. However, given a doubly stochastic matrix $\mathcal{P}$, the Birkhoff-von Neumann decomposition (Birkhoff, 1946) can be used to compute *some* distribution $\pi$ consistent with $\mathcal{P}$, i.e., $\mathcal{P}^{(\pi)} = \mathcal{P}$; any consistent distribution $\pi$ will suffice for our purposes.

## 2.2 Merit, Uncertainty, and Fairness

The principal must determine a distribution over rankings of the agents. This distribution will be based on some evidence for the agents' merits. This evidence could take the form of star ratings and reviews of products (combined with the site visitor's partially known preferences), or GPA, test scores, and recommendation letters of an applicant. Our main departure from past work on individual fairness (following (Dwork et al., 2012)) is that we do not view this evidence as having inherent meaning; rather, its sole role is to induce a posterior joint distribution over the agents' merits.

The merit of agent $x$ is $v_x \in \mathbb{R}$, and we write $\boldsymbol{v} = (v_x)_{x \in \mathcal{X}}$ for the vector of all agents' merits. Based on all observed evidence, the principal can infer a distribution $\Gamma$ over agents' merits using any suitable Bayesian inference procedure. Since the particular Bayesian model depends on the application, for our purposes, we merely assume that a posterior distribution $\Gamma$ was inferred using best practices and that ideally, this model is open to verification and audit.

We write $\Gamma(\boldsymbol{v})$ for the probability of merits $\boldsymbol{v}$ under $\Gamma$. We emphasize that the distribution will typically not be independent over entries of $\boldsymbol{v}$ — for example, students' merit conditioned on observed grades will be correlated via common grade inflation if they took the same class. To avoid awkward tie-breaking issues, we assume that $v_x \neq v_y$ for all distinct $x, y \in \mathcal{X}$ and all $\boldsymbol{v}$ in the support of $\Gamma$. This side-steps having to define the notion of top-$k$ lists with ties, and comes at little cost in expressivity, as any tie-breaking would typically be encoded in slight perturbations to the $v_x$ anyway.

We write $\mathcal{M}_{x,k}^{(\boldsymbol{v})}$ for the event that under $\boldsymbol{v}$, agent $x$ is among the top $k$ agents with respect to merit, i.e., that $|\{x' \mid v_{x'} > v_x\}| < k$. We now come to our key definition of approximate fairness.

**Definition 2.1** (Approximately Fair Ranking Distribution). A ranking distribution $\pi$ is $\phi$-*fair* iff

$$\sum_{k'=1}^{k} p_{x,k'}^{(\pi)} \geq \phi \cdot \mathbb{P}_{\boldsymbol{v} \sim \Gamma} \left[ \mathcal{M}_{x,k}^{(\boldsymbol{v})} \right] \tag{1}$$

for all agents $x$ and positions $k$. That is, the ranking distribution $\pi$ ranks $x$ at position $k$ or above with at least a $\phi$ fraction of the probability that $x$ is actually among the top $k$ agents according to $\Gamma$. Furthermore, $\pi$ is *fair* iff it is 1-fair.

The reason for defining $\phi$-approximately fair ranking distributions (rather than just fair distributions) is that fairness typically comes at a cost to the principal (such as lower expected clickthrough or lower expected performance of recommended employees). For example, if the $v_x$ are probabilities that a user will purchase products on an e-commerce site, then deterministically ranking by decreasing $\mathbb{E}_\Gamma [v_x]$ is the principal's optimal ranking under common assumptions about user behavior; yet, being deterministic, it is highly unfair. Our definition of approximate fairness allows, e.g., a policymaker to choose a trade-off regarding how much fairness (with resulting expected utility loss) to require from the principal. Notice that for $\phi = 0$, the principal is unconstrained.

We remark that the merit values $v_x$ only matter insofar as comparison is concerned; in other words, they are used ordinally, not cardinally. This is captured by the following proposition.

**Proposition 2.1.** Let $f : \mathbb{R} \to \mathbb{R}$ be any strictly increasing function. Let $\Gamma'$ be the distribution that draws the vector $(f(v_x))_{x \in \mathcal{X}}$ with probability $\Gamma(\boldsymbol{v})$ for all $\boldsymbol{v}$; that is, it replaces each entry $v_x$ with $f(v_x)$. Then, a ranking distribution $\pi$ is $\phi$-fair with respect to $\Gamma$ if and only if it is $\phi$-fair with respect to $\Gamma'$.

*Proof.* Because $f$ is strictly increasing, $\mathbb{P}_{\boldsymbol{v} \sim \Gamma}\left[\mathcal{M}_{x,k}^{(\boldsymbol{v})}\right] = \mathbb{P}_{\boldsymbol{v} \sim \Gamma'}\left[\mathcal{M}_{x,k}^{(\boldsymbol{v})}\right]$ for all $x$ and $k$. This immediately implies the claim by examining Definition 2.1. $\qquad \square$

Proposition 2.1 highlights a key aspect of our fairness definition: we avoid expressing any notion of fairness when "one agent has just 'a little' more merit than the other," instead arguing that fairness is only truly violated when an agent with more merit is treated worse than one with less merit. In other words, fairness is inherently *ordinal* in our treatment. This viewpoint has implications for a principal seeking "high-risk high-reward" agents, which we discuss in more depth in § B.2.

## 2.3 The Principal's Utility

The principal's utility can be the profit of an e-commerce site or the satisfaction of its customers. We assume that the utility for a ranking $\sigma$ with agent merits $\boldsymbol{v}$ takes the form $U(\sigma \mid \boldsymbol{v}) = \sum_{k=1}^{n} w_k v_{\sigma(k)}$, where $w_k$ is the position weight for position $k$ in the ranking, and we assume that the $w_k$ are non-increasing, i.e., the principal derives the most utility from earlier positions of the ranking.

The assumption that the utility from each position is factorable (i.e., of the form $w_k \cdot v_{\sigma(k)}$) is quite standard in the literature (Järvelin and Kekäläinen, 2002; Taylor et al., 2008). The assumption that the utility is *linear* in $v_{\sigma(k)}$ is in fact not restrictive at all. To see this, assume that the principal's utility were of the form $w_k \cdot f(v_{\sigma(k)})$ for some strictly increasing function $f$. By Proposition 2.1, the exact same fairness guarantees are achieved when the agents' merits $v_x$ are replaced with $f(v_x)$; doing so preserves fairness, while in fact making the principal's utility linear in the merits. Some common examples of utility functions falling into this general framework are DCG with $w_k = 1/\log_2(1 + k)$, Average Reciprocal Rank with $w_k = 1/k$, and Precision@K with $w_k = \mathbb{1}[k \leq K]/K$.

When the ranking and merits are drawn from distributions, the principal's utility is the expected utility under both sources of randomness:

$$U(\pi \mid \Gamma) = \mathbb{E}_{\sigma \sim \pi, \boldsymbol{v} \sim \Gamma}\left[U(\sigma \mid \boldsymbol{v})\right]. \tag{2}$$

## 2.4 Discussion

Our definition is superficially similar to existing definitions of individual fairness (e.g., (Dwork et al., 2012; Joseph et al., 2016)), in that similar observable features often lead to similar outcomes. Importantly, though, it side-steps the need to define a similarity metric between agents in the feature space. Furthermore, it does not treat the observable attributes (such as star ratings) themselves as any notion of "merit." Instead, our central point is that agents' features should be viewed *solely* as noisy signals about the agents' merits and that a comparison of their merits — and the principal's uncertainty about the merits — should determine the agents' relative ranking. That moves the key task of quantifying individual fairness from articulating which features should be considered relevant for similarity, to articulating what inferences can be drawn about merit from observed features.

**Merit as an Abstraction Boundary between Data and Fairness.** One may argue, rightfully, that from an operational perspective, our approach simply pushes the normative decisions into determining $\Gamma$. For example, if the distribution $\Gamma$ were biased in favor of or against a particular group, then the decisions of a supposedly fair algorithm (with respect to $\Gamma$) would in fact be unfair to that group. However, our main point is that normative decisions *should* indeed be encoded in the distribution $\Gamma$. To appreciate the conceptual approach, first notice that any algorithm implicitly encodes normative decisions, merely in the outputs it produces, which will favor some agents over others. The key question is how these normative decisions are encoded, how they can be articulated, and whether they could possibly be audited. If they are encoded in ad hoc algorithmic choices, articulating and auditing them may be difficult.

As an example, consider an admissions officer at a university, who believes that the GPA or SAT scores of affluent applicants may be higher due to access to tutors, rather than true academic potential. One approach to compensate for this advantage could be to subtract some (wealth-dependent) amount from an applicant's scores. A more principled approach — and the one we advocate — is for the admissions officer to explicitly express the possible distribution of merits given the test scores and wealth. The suitable notion of fairness is then derived by our framework, rather than as an ad hoc choice. A substantive discussion can then be had around the assumptions that go into the admission officer's particular choice of distribution, whether a different distribution would be more suitable, etc.

In a sense, the notion of merit, and distributions thereof, serves as a clean abstraction boundary between available data, and the desired fairness and utility. We argue that frequently, the difficult question to address is not so much what is "fair," but what the data truly reveal about an agent's merit. The latter should be articulated by domain experts, whereas the role of computer science is to provide frameworks for deriving fair algorithms *given* the merit distributions, as well as statistical approaches that may guide the derivation of $\Gamma$ from data.

**Randomization, Fairness, and Single-Shot Scenarios.**   In the introduction, we discussed two possible applications in which fairness is desirable: ranking of products in online e-commerce, and ranking of job applicants. We note that these two settings differ along an important dimension: e-commerce sites typically display/rank the same set of products many times over a short period of time, and the stakes each time are fairly low. On the other hand, any one particular job is a one-shot setting with high stakes. Intuitively, it "feels" like the use of randomization as a means to achieve fairness is more natural in the former setting than the latter. We discuss this issue in more depth.

First, we consider two practical reasons for randomization being more natural in repeated low-stakes settings: (1) In a high-stakes situation, a principal may be less willing to trade off utility for fairness, and (2) If fairness is *required* of the principal (rather than the principal's own goal), in a single-shot setting, it is much more difficult to verify that a decision was indeed made probabilistically; in contrast, for a repeated setting, statistical tools can be employed to keep the principal honest.

More fundamentally, the two settings differ in the point in time at which fairness is guaranteed. For concreteness, consider the simplest setting: two agents with identical posterior distributions vie for one position. In this case, a coin flip is *ex ante* fair: before the coin flip is realized, both agents have the same probability of being selected. However, it is not fair *ex post*: despite both having equal merit, one was selected, and the other was not. Contrast this with the alternative in which the same two agents compete multiple times, and a coin is flipped each time. Ex ante fairness is of course preserved, but even ex post, each agent was selected approximately the same number of times. This example may explain why randomization "feels" more fair for repeated than one-shot settings.

The fact that for a one-shot setting, a coin flip is only ex ante fair, however, does not obviate the need for making fair decisions in one-shot settings. There will be situations in which a principal is faced with multiple essentially indistinguishable agents and not enough positions for all of them.[1] While ex ante fairness may not be completely satisfactory, it still guarantees "more" fairness than arbitrary deterministic tie-breaking. Indeed, one may argue that the goal of many principals is not so much to make fair decisions as to make defensible ones. For example, if applicants are ranked strictly by GPA, choosing an applicant with GPA of 3.91 over one with GPA of 3.90 is essentially random tie-breaking, but with a rule that can be defended. Our point here is that randomization should be considered as a viable alternative, if the true goal is to achieve fairness.

**Other Considerations.**   In extending the probabilistic fairness axiom from two to multiple agents in Equation (1), we chose to axiomatize fairness in terms of which position agents are assigned to. An equally valid generalization would have been to require for each pair $x, y$ of agents that if $x$ has more merit than $y$ with probability at least $\rho$, then $x$ must precede $y$ in the ranking with probability at least $\phi \cdot \rho$. The main reason why we prefer Equation (1) is computational: the only linear programs we know for the alternative approach require variables for all rankings and are thus exponential (in $n$) in size. Exploring the alternative definition is an interesting direction for future work.

Due to space constraints, additional properties of our fairness definition are discussed in § B of the supplementary material. In particular, we discuss how fairness requirements give the principal

---

[1]For example, in college admissions, qualified applicants typically outnumber available slots, and differences among the qualified applicants are frequently very small.

stronger incentives for obtaining more accurate estimates of agents' merits, and how to align the ordinal nature of our definition with a principal's interest in selecting high-risk high-reward agents.

# 3  Optimal and Fair Policies

For a distribution $\Gamma$ over merits, let $\sigma_\Gamma^*$ be the ranking which sorts the agents by expected merit, i.e., by non-increasing $\mathbb{E}_{\boldsymbol{v} \sim \Gamma}[v_x]$. The following well-known proposition follows because the position weights $w_k$ are non-increasing.

**Proposition 3.1.** $\sigma_\Gamma^*$ is a utility-maximizing ranking policy for the principal.

If the principal's expected utility can be evaluated efficiently, computing $\sigma_\Gamma^*$ only requires sorting the agents by utility, and thus takes time only $O(n \log n)$. While this policy conforms to the Probability Ranking Principle (Robertson, 1977), it violates Axiom 1 for ranking fairness when merits are uncertain. We define a natural solution for a 1-fair ranking distribution based on Thompson Sampling:

**Definition 3.1** (Thompson Sampling Ranking Distribution). Define $\pi_\Gamma^{\mathrm{TS}}$ as follows: first, draw a vector of merits $\boldsymbol{v} \sim \Gamma$, then rank the agents by decreasing merits in $\boldsymbol{v}$.

That $\pi_\Gamma^{\mathrm{TS}}$ is 1-fair follows directly from the definition of fairness. By definition, it ranks each agent $x$ in position $k$ with exactly the probability that $x$ has $k$-th highest merit.

**Proposition 3.2.** $\pi_\Gamma^{\mathrm{TS}}$ is a 1-fair ranking distribution.

Furthermore, computing $\pi_\Gamma^{\mathrm{TS}}$ only involves sampling from $\Gamma$ and then sorting the agents by merit, so it can be efficiently performed in time $O(n \log n)$.

## 3.1  Trading Off Utility and Fairness

One straightforward way of trading off between the two objectives of fairness and principal's utility is to randomize between the two policies $\pi_\Gamma^{\mathrm{TS}}$ and $\pi^*$.

**Definition 3.2** (OPT/TS-Mixing). The OPT/TS-Mixing ranking policy $\pi^{\mathrm{Mix},\phi}$ randomizes between $\pi_\Gamma^{\mathrm{TS}}$ and $\pi_\Gamma^*$ with probabilities $\phi$ and $1 - \phi$, respectively.

This policy inherits a runtime of $O(n \log n)$ from $\pi_\Gamma^{\mathrm{TS}}$ and $\pi_\Gamma^*$. The following lemma gives guarantees for such randomization (but we will later see that this strategy is suboptimal).

**Lemma 3.1.** Consider two ranking policies $\pi_1$ and $\pi_2$ such that $\pi_1$ is $\phi_1$-fair and $\pi_2$ is $\phi_2$-fair. A policy that randomizes between $\pi_1$ and $\pi_2$ with probabilities $q$ and $1 - q$, respectively, is at least $(q\phi_1 + (1 - q)\phi_2)$-fair and obtains expected utility $qU(\pi_1 \mid \Gamma) + (1 - q)U(\pi_2 \mid \Gamma)$.

**Corollary 3.1.** The ranking policy $\pi^{\mathrm{Mix},\phi}$ is $\phi$-fair.

By definition, $\pi^{\mathrm{Mix},\phi=0}$ has the highest utility among all 0-fair ranking policies. Furthermore, all 1-fair policies achieve the same utility since the fairness axiom for $\phi = 1$ completely determines the marginal rank probabilities.

**Lemma 3.2.** All 1-fair ranking policies have the same utility for the principal.

While $\pi^{\mathrm{Mix},\phi=0}$ and $\pi^{\mathrm{Mix},\phi=1}$ have the highest utility among 0-fair and 1-fair ranking policies, respectively, $\pi^{\mathrm{Mix},\phi}$ will typically not have maximum utility among all $\phi$-fair ranking policies for other values of $\phi \in (0, 1)$. We illustrate this with a concrete example with $n = 3$ agents in § D.

## 3.2  Optimizing Utility for $\phi$-Fair Rankings

We now formulate a linear program for computing the policy $\pi^{\mathrm{LP},\phi}$ that maximizes the principal's utility, subject to being $\phi$-fair. The variables of the linear program are the marginal rank probabilities $p_{x,k}^{(\pi)}$ of the distribution $\pi$ to be determined. Then, by Equation (2) and linearity of expectation, the principal's expected utility can be written as $U(\pi \mid \Gamma) = \sum_{x \in \mathcal{X}} \sum_k p_{x,k}^{(\pi)} \cdot \mathbb{E}_{\boldsymbol{v} \sim \Gamma}[v_x] \cdot w_k$. We use this linear form of the utilities to write the optimization problem as the following LP with variables

$p_{x,k}$ (omitting $\pi$ from the notation):

$$
\begin{aligned}
\text{Maximize} \quad & \sum_x \sum_k p_{x,k} \cdot \mathbb{E}_{\boldsymbol{v} \sim \Gamma}\left[v_x\right] \cdot w_k \\
\text{subject to} \quad & \sum_{k'=1}^{k} p_{x,k'} \geq \phi \cdot \mathbb{P}_{\boldsymbol{v} \sim \Gamma}\left[\mathcal{M}_{x,k}^{(\boldsymbol{v})}\right] && \text{for all } x, k \\
& \sum_{k=1}^{n} p_{x,k} = 1 && \text{for all } x \\
& \sum_x p_{x,k} = 1 && \text{for all } k \\
& 0 \leq p_{x,k} \leq 1 && \text{for all } x, k.
\end{aligned}
\tag{3}
$$

In the LP, the first set of constraints captures $\phi$-approximate fairness for all agents and positions, while the remaining constraints ensure that the marginal probabilities form a doubly stochastic matrix.

As a second step, the algorithm uses the Birkhoff-von Neumann (BvN) Decomposition of the matrix $\mathcal{P} = (p_{x,k})_{x,k}$ to explicitly obtain a distribution $\pi$ over rankings such that $\pi$ has marginals $p_{x,k}$. The Birkhoff-von Neumann Theorem (Birkhoff, 1946) states that the set of doubly stochastic matrices is the convex hull of the permutation matrices, which means that we can write $\mathcal{P} = \sum_\sigma q_\sigma \mathcal{P}^{(\sigma)}$, where $\mathcal{P}^{(\sigma)}$ is the binary permutation matrix corresponding to the deterministic ranking $\sigma$, and the $q_\sigma$ form a probability distribution. It was already shown by Birkhoff (1946) how to find a polynomially sparse decomposition in polynomial time.

Having to solve a linear program obviously makes the computation of $\pi^{\mathrm{LP},\phi}$ less efficient. It is still efficient enough to be feasible for several hundred agents. An interesting direction for future work would be whether the specific LP can be solved more efficiently, either exactly or approximately, by using algorithms other than the standard ones (Ellipsoid or Interior Point Methods).

In order to solve the Linear Program (3), one needs to know $\mathbb{P}_{\boldsymbol{v} \sim \Gamma}\left[\mathcal{M}_{x,k}^{(\boldsymbol{v})}\right]$ for all $i$ and $k$. For some distributions $\Gamma$ (e.g., Example 2), these quantities can be calculated in closed form. For others, they can be approximated using Monte Carlo sampling. Small approximation errors only have a small impact on the final solution as captured by Propositions C.1 and C.2 in the appendix.

## 4   Experimental Evaluation: MovieLens Dataset

To evaluate our approach in a recommendation setting with a realistic preference distribution, we designed the following experimental setup based on the MovieLens 100K (ML-100K) dataset. The dataset contains 100,000 ratings, by 600 users, on 9,000 movies belonging to 18 genres (Harper and Konstan, 2015). In our setup, for each user, the principal is a recommender system that has to generate a ranking of movies $\mathcal{S}_g$ for one of the genres $g$ (e.g., Horror, Romance, Comedy) according to a notion of *merit* of the movies we define as follows.

**Modeling the Merit Distribution.**   We define the (unknown) merit $v_m$ of a movie $m$ as the average rating of the movie across the user population[2] — this merit is unknown because most users have not seen/rated most movies. To be able to estimate this merit based on ratings in the ML-100K dataset, and to concretely define its underlying distribution and the corresponding fairness criteria, we define a generative model of user ratings. The model assumes that the rating of a movie $m \in \mathcal{S}_g$ is drawn from a multinomial distribution over $\{1, 2, 3, 4, 5\}$ with parameters $\boldsymbol{\theta}_m = (\theta_{m,1}, \ldots, \theta_{m,5})$. Assuming a Dirichlet prior, we can infer the posterior distribution of the ratings (and hence the merit) from the dataset in closed form, using merely the counts of each rating for each movie in the dataset. (See § E.1 for details.) Based on this definition of merit and its uncertainty, we define and compare different ranking policies for this experimental setup as follows.

**Utility Maximizing Ranking** ($\pi^*$): We use the DCG function (Burges et al., 2005) with position weights $w_k = 1/\log_2(1+k)$ as our utility measure. Since the weights are indeed strictly decreasing, as described in § 3, the optimal ranking policy $\pi^*$ sorts the movies (for the particular query) by decreasing expected merit, which is the expected average rating $\overline{v}_m \triangleq \mathbb{E}_{\boldsymbol{\theta} \sim \mathbb{P}[\boldsymbol{\theta}_m \mid \mathcal{D}]}\left[v_m(\boldsymbol{\theta})\right]$ under the posterior Dirichlet distribution in our case; here, $v_m(\boldsymbol{\theta})$ is the average rating of movie $m$ corresponding to the parameter vector $\boldsymbol{\theta}$ sampled from the posterior.

---

[2]For a personalized ranking application, an alternative would be to choose each user's (mostly unknown) rating as the merit criterion instead of the average rating across the user population.

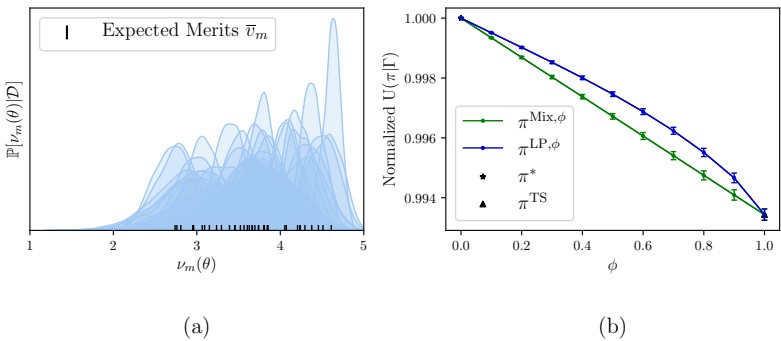

(a)                                          (b)

Figure 1: (a) Posterior distribution of ratings (merit) for a subset of "Comedy" movies, (b) Tradeoff between Utility and Fairness, as captured by $\phi$.

**Fully Fair Ranking Policy** ($\pi^{\text{TS}}$): A fair ranking, in this case, ensures that, for all positions $k$, a movie is placed in the top $k$ positions according to the posterior merit distribution. In this setup, a fully fair ranking policy $\pi^{\text{TS}}$ is obtained by sampling the multinomial parameters $\boldsymbol{\theta}_m$ for each movie $m \in \mathcal{S}_g$ and computing $v_m(\boldsymbol{\theta}_m)$ to rank them:

$$\pi^{\text{TS}}(\mathcal{S}_g) \sim \text{argsort}_m \, v_m(\boldsymbol{\theta}_m) \text{ s.t. } \boldsymbol{\theta}_m \sim \mathbb{P}[\boldsymbol{\theta}_m | \mathcal{D}].$$

**OPT/TS-Mixing Ranking Policy** ($\pi^{\text{Mix},\phi}$): The policies $\pi^{\text{Mix},\phi}$ randomize between the fully fair and utility-maximizing ranking policies with probabilities $\phi$ and $1 - \phi$, respectively.

**LP Ranking Policy** ($\pi^{\text{LP},\phi}$): The $\phi$-fair policies $\pi^{\text{LP},\phi}$ require the principal to have access to the probabilities $\mathbb{P}_{\boldsymbol{v} \sim \Gamma}\left[\mathcal{M}_{m,k}^{(\boldsymbol{v})}\right]$ which we estimate using $5 \cdot 10^4$ Monte Carlo samples, so that any estimation error becomes negligible.

**Observations and Results.** In the experiments presented, we used the ranking policies $\pi^*$, $\pi^{\text{TS}}$, $\pi^{\text{Mix},\phi}$ and $\pi^{\text{LP},\phi}$ to create separate rankings for each of the 18 genres. For each genre, the task is to rank a random subset of 40 movies from that genre. To get a posterior with an interesting degree of uncertainty, we take a 10% i.i.d. samples from $\mathcal{D}$ to infer the posterior for each movie. We observe that the results are qualitatively consistent across genres, and we thus focus on detailed results for the genre "Comedy" as a representative example. Its posterior merit distribution over a subset is visualized in Figure 1(a). Note that substantial overlap exists between the marginal merit distributions of the movies, indicating that as opposed to $\pi^*$ (which sorts based on the expected merits), the policy $\pi^{\text{TS}}$ will randomize over many different rankings.

**Observation 1:** We evaluate the cost of fairness to the principal in terms of loss in utility, as well as the ability of $\pi^{\text{LP},\phi}$ to minimize this cost for $\phi$-fair rankings. Figure 1(b) shows this cost in terms of expected Normalized DCG (i.e., $\text{NDCG} = \text{DCG}/\max(\text{DCG})$ as in (Järvelin and Kekäläinen, 2002)). These results are averaged over 20 runs with different subsets of movies and different training samples. The leftmost end corresponds to the NDCG of $\pi^*$, while the rightmost point corresponds to the NDCG of the 1-fair policy $\pi^{\text{TS}}$.

The drop in NDCG is below one percentage point, which is consistent with the results for the other genres. We also conducted experiments with other values of $s$, data set sizes, and choices of $w_k$; even under the most extreme conditions, the drop was at most 2 percent. While this rather small drop may be surprising at first, we point out that uncertainty in the estimates affects the utility of both $\pi^*$ and $\pi^{\text{TS}}$. By industry standards, a 2% drop in NDCG is considered quite substantial; however, it is not catastrophic and hence bodes well for possible adoption.

**Observation 2:** Figure 1(b) also compares the trade-off in NDCG in response to the fairness approximation parameter $\phi$ for both $\pi^{\text{Mix},\phi}$ and $\pi^{\text{LP},\phi}$. We observe that the utility-optimal policy $\pi^{\text{LP},\phi}$ provides gains over $\pi^{\text{Mix},\phi}$, especially for large values of $\phi$. Thus, using $\pi^{\text{LP},\phi}$ can further reduce the cost of fairness discussed above.

**Observation 3:** To provide intuition about the difference between $\pi^{\text{Mix},\phi}$ and $\pi^{\text{LP},\phi}$, Figure 4 in § E.3 visualizes the marginal rank distributions $p_{m,k}$, i.e., the probability that movie $m$ is ranked

at position $k$. The key distinction is that, for intermediate values of $\phi$, $\pi^{\mathrm{LP},\phi}$ exploits a non-linear structure in the ranking distribution (to achieve a better trade-off) while $\pi^{\mathrm{Mix},\phi}$ merely interpolates linearly between the solutions for $\phi = 0$ and $\phi = 1$.

Based on these observations, in general, the utility loss due to fairness is small, and can be further reduced by optimizing the ranking distribution with the LP-based approach. These results are based on the definition of merit as the average rating of movies over the entire user population. A more realistic setting would personalize rankings for each user, with merit defined as the expected relevance of a movie to the user. In our experiments, the results under such a setup were quite similar, and are hence omitted for brevity and clearer illustration.

# 5  Real-World Experiment: Paper Recommendation

To study the effect of deploying a fair ranking policy in a real ranking system, we built and fielded a paper recommendation system at the 2020 ACM SIGKDD Conference on Knowledge Discovery and Data Mining. The goal of the experiment is to understand the impact of fairness under real user behavior, as opposed to simulated user behavior that is subject to modeling assumptions. Specifically, we seek to answer two questions: (a) Does a fair ranking policy lead to a more equitable distribution of exposure among the papers? (b) Does fairness substantially reduce the utility of the system to the users?

The users of the paper recommendation system were the participants of the conference, which was held virtually in 2020. Signup and usage of the system was voluntary. Each user was recommended a personalized ranking of the papers published at the conference. This ranking was produced either by $\sigma^*$ or by $\pi^{\mathrm{TS}}$, and the assignment of users to treatment ($\pi^{\mathrm{TS}}$) or control ($\sigma^*$) was randomized.

**Modeling the Merit Distribution.**    The merit of a paper for a particular user is based on a relevance score $\mu_{u,i}$ that relates features of the user (e.g., bag-of-words representation of recent publications, co-authorship) to features of each conference paper (e.g., bag-of-words representation of paper, citations). Most prominently, the relevance score $\mu_{u,i}$ contains the TFIDF-weighted cosine similarity between the bag-of-words representations.

To model the uncertainty in $\mu_{u,i}$, we make the assumption that since all papers were accepted to the conference, they must have been deemed relevant to at least some fraction of the audience by the peer reviewers. Hence, papers with uniformly low relevance scores $\mu_{u,i}$ across users (e.g., ones introducing new research directions or bringing in novel techniques) must have a higher uncertainty in their relevance score. This assumption leads us to define the uncertainty as a normal distribution around the mean $\mu_{u,i}$ with standard deviation $\delta_i$ defined such that, for paper $i$, there exists at least one user who finds the paper highly relevant to their interests with high probability. (See § F.2 for details on how $\delta_i$ is calculated.)

**Ranking Policies.**    Users in the control group $\mathcal{U}_{\pi^*}$ received rankings in decreasing order of $\mu_{u,i}$. Users in the treatment group $\mathcal{U}_{\pi^{\mathrm{TS}}}$ received rankings from the fair policy that sampled scores from the uncertainty distribution, $\hat{\mu}_{u,i} \sim \mathcal{N}(\mu_{u,i}, \delta_i)$, and ranked the papers by decreasing $\hat{\mu}_{u,i}$.

**Results and Observations.**    We first analyze if the fair policy provided more equitable exposure to the papers. In this real-world evaluation, exposure is not equivalent to rank, but depends on whether users actually browsed to a given position in the ranking. Users could browse their ranking in pages of 5 recommendations each; we count a paper as *exposed* if the user scrolled to its page.

**Observation 1:** Figure 2 compares the histograms of exposure of the papers in the treatment and control groups. Under the fair policy, the number of papers in the lowest tier of exposure is roughly halved compared to the control condition. This verifies that the fair policy does have an impact on exposure in a real-world setting, and it aligns with our motivation that a fair ranking policy distributes exposure more equally among the ranked agents. This is also supported by comparing the Gini inequality coefficient (Gini, 1936) for the two distributions: $G(\pi^*) = 0.3302$, while $G(\pi^{\mathrm{TS}}) = 0.2996$ (where a smaller coefficient means less inequality in the distribution).

**Observation 2:** To evaluate the impact of fairness on user engagement, we analyze a range of engagement metrics as summarized in Table 1. While a total of 923 users signed up for the system ahead of the conference (and were randomized into treatment and control groups), 462 never came to

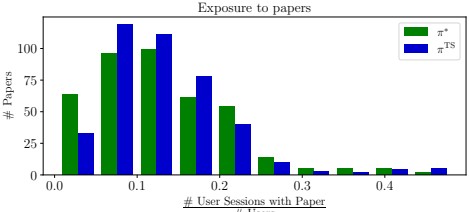

Figure 2: Distribution of the exposure of papers in treatment and control.

| | Number of Users with activity | | Average Activity Per User | |
|---|---|---|---|---|
| | $\pi^*$ | $\pi^{\text{TS}}$ | $\pi^*$ | $\pi^{\text{TS}}$ |
| Total Number of users | 213 | 248 | - | - |
| Num. of pages examined | - | - | 10.8075 | 10.7984 |
| Read Abstract | 92 | 101 | 3.7230 | 2.6774 |
| Add to Calendar | 51 | 50 | 1.4366 | 0.8508 |
| Read PDF | 40 | 52 | 0.5258 | 0.5323 |
| Add Bookmark | 16 | 13 | 0.3192 | 0.6129 |

Table 1: User engagement under the two conditions $\pi^*$ and $\pi^{\text{TS}}$. None of the differences are statistically significant. (For user actions, this is specifically due to the small sample size).

the system after all. Of the users that came, 213 users were in $\mathcal{U}_{\pi^*}$, and 248 users were in $\mathcal{U}_{\pi^{\text{TS}}}$. Note that this difference is not caused by the treatment assignment, since users had no information about their assignment/ranking before entering the system. The first engagement metric we computed is the average number of pages that users viewed under both conditions. With roughly 10.8 pages (about 54 papers), engagement under both conditions was almost identical. Users also had other options to engage, but there is no clear difference between the conditions, either. On average, they read more paper abstracts and added more papers to their calendar under the control condition, but read more PDF and added more bookmarks under the treatment condition. However, none of these differences is significant at the 95% level for either a Mann-Whitney U test or a two-sample t-test. While the sample size is small, these findings align with the findings on the synthetic data, namely that fairness did not appear to place a large cost on the principal (here representing the users).

## 6 Conclusions and Future Work

We believe that the focus on uncertainty we proposed in this paper constitutes a principled approach to capturing the intuitive notion of fairness to agents with similar features: rather than focusing on the features themselves, the key insight is that the features' similarity entails significant statistical uncertainty about which agent has more merit. Randomization provides a way to fight fire with fire, and axiomatize fairness in the presence of such uncertainty.

Our work raises a wealth of questions for future work. Perhaps most importantly, as discussed in § 2.4, to operationalize our proposed notion of fairness, it is important to derive principled merit distributions $\Gamma$ based on the observed features. Our experiments were based on "reasonable" notions of merit distributions and concluded that fairness might not have to be very expensive to achieve for a principal. However, much more experimental work is needed to truly evaluate the impact of fair policies on the utility that is achieved. It would be particularly intriguing to investigate which types of real-world settings lend themselves to implementing fairness at little cost, and which force a steep trade-off between the two objectives.

Our work also raises several interesting theoretical questions. In § B.1, we show one setting in which forcing the principal to use a fair policy drastically increases the principal's incentives to form a more accurate posterior $\Gamma$ for a minority group. We did not prove a general result in this vein. We ask: will the incentives of a principal to learn a better posterior $\Gamma$ always (weakly) increase if the principal is forced to be fairer? If true, this would provide a fascinating additional benefit of fairness requirements.

Another interesting question concerns the utility loss incurred by using the policy OPT/TS-Mixing. As shown in § D, OPT/TS-Mixing is in general not optimal. However, in the example from § D as well as in our experiments in § 4, the loss in utility was quite small. An interesting question would be to bound the worst-case loss in the utility of OPT/TS-Mixing, compared to the LP-based policy. In particular, this question is of interest due to the simplicity of the OPT/TS-Mixing policy; it does not require the computationally expensive solution of an LP or an explicit estimate of marginal rank probabilities under $\Gamma$.

## Acknowledgements

This research was supported in part by NSF Award IIS-2008139 and IIS-1901168. We thank Aleksandra Korolova, Cris Moore, and Stephanie Wykstra for useful discussions and feedback. Any opinions, findings, and conclusions or recommendations expressed in this material are those of the author(s) and do not necessarily reflect the views of the National Science Foundation.

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
