# Supplementary Material: Fairness in Ranking under Uncertainty

## A    Related Work

As algorithmic techniques, especially machine learning, find widespread applications in decision making, there is notable interest in understanding its societal impacts. While algorithmic decisions can counteract existing biases by preventing human error and implicit bias, data-driven algorithms may also create new avenues for introducing unintended bias (Barocas and Selbst, 2016). There have been numerous attempts to define notions of fairness in the supervised learning setting, especially for binary classification and risk assessment (Calders et al., 2009; Zliobaite, 2015; Dwork et al., 2012; Hardt et al., 2016; Mehrabi et al., 2019). The group fairness perspective imposes constraints like demographic parity (Calders et al., 2009; Zliobaite, 2015) and equalized odds (Hardt et al., 2016). Follow-up work has proposed techniques for implementing fairness through pre-processing methods (Calmon et al., 2017; Lum and Johndrow, 2016), in process while learning the model (Zemel et al., 2013; Woodworth et al., 2017; Zafar et al., 2017) and post-processing methods (Hardt et al., 2016; Pleiss et al., 2017; Kim et al., 2019), in addition to causal approaches to fairness (Kilbertus et al., 2017; Kusner et al., 2017).

Individual fairness, on the other hand, is concerned with comparing the outcomes of agents directly, not in aggregate. Specifically, the individual fairness axiom states that two individuals similar with respect to a task should receive similar outcomes (Dwork et al., 2012). While the property of individual fairness is highly desirable, it is hard to define precisely; in particular, it is highly dependent on the definition of a suitable similarity notion. Although similar in spirit, our work sidesteps this need to define a similarity metric between agents in the feature space. Rather, we view an agent's features solely as noisy signals about the agent's merit and posit that a comparison of these merits — and the principal's uncertainty about them — should determine the relative ranking. Individual fairness definitions have also been adopted in online learning settings such as stochastic multi-armed bandits (Patil et al., 2020; Heidari and Krause, 2018; Schumann et al., 2019; Celis et al., 2018), where the desired property is that a worse arm is never "favored" over a better arm despite the algorithm's uncertainty over the true payoffs (Joseph et al., 2016), or a smooth fairness assumption that a pair of arms be selected with similar probability if they have a similar payoff distribution (Liu et al., 2017). While these definitions are derived from the same tenet of fairness as Axiom 1 for a pair of agents, we extend it to rankings, where $n$ agents are compared at a time.

Rankings are a primary interface through which machine learning models support human decision making, ranging from recommendation and search in online systems to machine-learned assessments for college admissions and recruiting. One added difficulty with considering fairness in the context of rankings is that the decision for an agent (where to rank that agent) depends not only on their own merits, but on others' merits as well (Dwork et al., 2019). The existing work can be roughly categorized into three groups: Composition-based, opportunity-based, and evidence-based notions of fairness. The notions of fairness based on the composition of the ranking operate along the lines of demographic parity (Zliobaite, 2015; Calders et al., 2009), proposing definitions and methods that minimize the difference in the (weighted) representation between groups in a prefix of the ranking (Yang and Stoyanovich, 2017; Celis et al., 2018; Asudehy et al., 2019; Zehlike et al., 2017; Mehrotra et al., 2018; Zehlike and Castillo, 2020). Other works argue against the winner-take-all allocation of economic opportunity (e.g., exposure, clickthrough, etc.) to the ranked agents or groups of agents, and that the allocation should be based on a notion of merit (Singh and Joachims, 2018; Biega et al., 2018; Diaz et al., 2020). Meanwhile, the metric-based notions equate a ranking with a set of pairwise comparisons, and define fairness notions based on parity of pairwise metrics within and across groups (Kallus and Zhou, 2019; Beutel et al., 2019; Narasimhan et al., 2020; Lahoti et al., 2019). Similar to pairwise accuracy definitions, evidence-based notions such as (Dwork et al., 2019) propose semantic notions such as *domination-compatibility* and *evidence-consistency*, based on relative ordering of subsets within the training data. Our fairness axiom combines the opportunity-based and evidence-based notions by stating that the economic opportunity allocated to the agents must be consistent with the existing evidence about their relative ordering.

Ranking has been widely studied in the field of Information Retrieval (IR), mostly in the context of optimizing user utility. The *Probability Ranking Principle (PRP)* (Robertson, 1977), a guiding principle for ranking in IR, states that user utility is optimal when documents (i.e., the agents) are ranked by expected values of their estimated relevance (merit) to the user. While this certainly holds

when the estimates are unbiased and devoid of uncertainty, we argue that it leads to unfair rankings for agents about whose merits the model might be uncertain. While the research on diversified rankings in IR appears related, in comparison to our work, the goal there is to maximize user utility alone by handling uncertainty about the user's information needs (Radlinski et al., 2009) and to avoid redundancy in the ranking (Clarke et al., 2008; Carbonell and Goldstein, 1998). Besides ranking diversity, IR methods have dealt with uncertainty in relevance that comes via users' implicit or explicit feedback (Penha and Hauff, 2021; Soufiani et al., 2012), as well as stochasticity arising from optimizing over probabilistic rankings instead of discrete combinatorial structures (Taylor et al., 2008; Burges et al., 2005). It is only recently that there has been an interest in developing evaluation metrics (Diaz et al., 2020) and learning algorithms (Singh and Joachims, 2019; Morik et al., 2020) that use stochastic ranking models to deal with unfair exposure.

Additional recent strands of work on fairness in selection problems focus on fairly selecting individuals distributed across different groups in the presence of group-based implicit bias (Kleinberg and Raghavan, 2018; Celis et al., 2020), noisy sensitive attributes (Mehrotra and Celis, 2021), or incomparable merits across different groups (Kearns et al., 2017). Kearns et al. (2017) present a way to fairly select $k$ individuals distributed across $d$ populations where each population can be sorted by merit without uncertainty but merit in one population cannot be directly compared to merit in another. Hence, they propose using the true CDF rank as a derived merit criterion that can be compared. There has also been recent interest in studying the effect of uncertainty regarding sensitive attributes, labels and other features used by the machine learning model on the accuracy-based fairness properties of the model (Ghosh et al., 2021; Prost et al., 2021). In contrast, our work takes a more fundamental approach to defining a merit-based notion of fairness arising due to the presence of uncertainty when estimating merits based on fully observed features and outcomes.

# B    Additional Model Discussion

## B.1    Information Acquisition Incentives for the Principal

An additional benefit of requiring the use of fair ranking policies is that it makes the principal bear more of the cost of an inaccurate $\Gamma$, and thereby incentivizes the principal to improve the distribution $\Gamma$. To see this at a high level, notice that if $\Gamma$ precisely revealed merits, then the optimal and fair policies would coincide. In the presence of uncertainty, an unrestricted principal will optimize utility, and in particular do better than a principal who is constrained to be (partially or completely) fair. Thus, a fair principal stands to gain more by obtaining perfect information. The following example shows that this difference can be substantial, i.e., the information acquisition incentives for a fair principal can be much higher.

**Example 1.** Consider again the case of a job portal. To keep the example simple, consider a scenario in which the portal tries to recommend exactly one candidate for a position.[3] There are two groups of candidates, which we call majority and underrepresented minority (URM). The majority group contains exactly one candidate of merit 1, all others having merit 0; the URM group contains exactly one candidate of merit $1 + \epsilon$, all others having merit 0 as well. Due to past experience with the majority group, the portal's distribution $\Gamma$ over merits precisely pinpoints the meritorious majority candidate, but reveals no information about the meritorious URM candidate; that is, the distribution places equal probability on each of the URM candidates having merit $1 + \epsilon$.

A utility-maximizing portal will therefore go with "the known thing," obtaining utility 1 from recommending the majority candidate. The loss in utility from ignoring the URM candidates is only $\epsilon$. Now consider a portal required to be 1-fair. Because each of the URM candidates is the best candidate with probability $1/n$ (when there are $n$ URM candidates), and the majority candidate is *known* to never be the best candidate, each URM candidate *must* be recommended with probability $1/n$. Here, the uncertainty about which URM candidate is meritorious will provide the portal with a utility that is only $(1+\epsilon)/n$.

In this example, fairness strengthens the incentive for the portal to acquire more information about the URM group; specifically, to learn to perfectly identify the meritorious candidate. Under full knowledge, the portal will now have utility $1 + \epsilon$ for both the fair and the utility-maximizing policy.

---

[3]This can be considered a ranking problem in which the first slot has $w_1 = 1$, while all other slots have weight 0.

For the utility-maximizing portal, this is the optimal choice; and for the fair strategy, it is perfectly fair to always select the (deterministically known) best candidate. Thus, a portal forced to use the fair strategy stands to increase its utility by a much larger amount; at least in this example, our definition of fairness splits the cost of a high-variance distribution $\Gamma$ more evenly between the principal and the affected agents when compared to the utility-optimizing policy, where almost all the cost of uncertainty is borne by the agents in the URM group. This drastically increases the principal's incentives for more accurate and equitable information gathering.

To what extent the insights from this example generalize to arbitrary settings (e.g., whether the principal *always* stands to gain more from additional information when forced to be fairer) is a fascinating direction for future research.

### B.2 Ordinal Merit and High-Risk High-Reward Agents

As we discussed earlier, Proposition 2.1 highlights the fact that our definition of fairness only considers ordinal properties, i.e., comparisons, of merit. This means that frequently selecting "moonshot" agents (those with very rare very high merit) would be considered unfair. We argue that this is not a drawback of our fairness definition; rather, if moonshot attempts are worth supporting frequently, then the definition of merit should be altered to reflect this understanding. As a result, viewing the merit definition under the prism of our fairness definition helps reveal misalignments between stated merit and actual preferences.

For a concrete example, consider two agents: agent A has known merit 1, while agent B has merit $M \gg 1$ with probability 1% and 0 with probability 99%. When $M > 100$, agent B has larger expected merit, but regardless of whether $M > 100$ or $M \leq 100$, a fully fair principal cannot select B with probability more than 1%. One may consider this a shortcoming of our model: it would prevent, for instance, a funding agency (which tries to be fair to research grant PIs) from focusing on high-risk high-reward research. We argue that the shortcoming will typically not be in the fairness definition, but in the chosen definition of merit. For concreteness, suppose that the status quo is to evaluate merit as the total number of citations which the funded work attracts during the next century.[4] Also, for simplicity, suppose that "high-reward" research is research that attracts more than 100,000 citations over the next century. If we consider one unit of merit as 1000 citations, and assume that the typical research grant results in work attracting about that many citations, then the funding agency faces the problem from the previous paragraph, and will not be able to support PI B with probability more than 1%. This goes against the express preference of many funding agencies for high-risk high-reward work.

However, if one truly believes that high-reward work is fundamentally different (e.g., it will change the world), then this difference should be explicitly modeled in the notion of merit. For example, rather than "number of citations," an alternative notion of merit would be "probability that the number of citations exceeds 100,000." This approach would allow the agency to select PIs based on the posterior probability (based on observed attributes, such as track record and the proposal) of producing such high-impact work. Of course, in reality, different aspects of merit can be combined to define a more accurate notion of merit that reflects what society values as true merit of research.

The restrictions imposed on a principal by our framework will and should force the principal to articulate actual merit of agents carefully, rather than adding ad hoc objectives. Once merit has been clearly defined, we anticipate that the conflict between fairness and societal objectives will be significantly reduced.

## C  Omitted Proofs

Here, we provide proofs omitted in § 3. The results are restated here for convenience. Proposition 3.1 is standard, and we only include a proof for completeness.

**Proposition 3.1**  $\sigma_\Gamma^*$ *is a utility-maximizing ranking policy for the principal, even over randomized policies.*

---

[4]This measure is chosen for simplicity of discussion, not to actually endorse this metric.

*Proof.* Let $\pi$ be a randomized policy for the principal. We will use a standard exchange argument to show that making $\pi$ more similar to $\sigma_\Gamma^*$ can only increase the principal's utility. Recall that by Equation (2), the principal's utility under $\pi$ can be written as

$$U(\pi \,|\, \Gamma) = \sum_{x \in \mathcal{X}} \sum_k p_{x,k}^{(\pi)} \cdot \mathbb{E}_{\boldsymbol{v} \sim \Gamma} [v_x] \cdot w_k.$$

Assume that $\pi$ does not sort $x$ by non-increasing $\mathbb{E}_{\boldsymbol{v} \sim \Gamma} [v_x]$. Then, there exist two positions $j < k$ and two agents $x, y$ such that $\mathbb{E}_{\boldsymbol{v} \sim \Gamma} [v_x] > \mathbb{E}_{\boldsymbol{v} \sim \Gamma} [v_y]$, and $p_{x,k}^{(\pi)} > 0$ and $p_{y,j}^{(\pi)} > 0$. Let $\epsilon = \min(p_{x,k}^{(\pi)}, p_{y,j}^{(\pi)}) > 0$, and consider the modified policy which subtracts $\epsilon$ from $p_{x,k}^{(\pi)}$ and $p_{y,j}^{(\pi)}$ and adds $\epsilon$ to $p_{x,j}^{(\pi)}$ and $p_{y,k}^{(\pi)}$. This changes the expected utility of the policy by

$$\epsilon \cdot (\mathbb{E}_{\boldsymbol{v} \sim \Gamma} [v_x] \cdot w_j + \mathbb{E}_{\boldsymbol{v} \sim \Gamma} [v_y] \cdot w_k - \mathbb{E}_{\boldsymbol{v} \sim \Gamma} [v_x] \cdot w_k - \mathbb{E}_{\boldsymbol{v} \sim \Gamma} [v_y] \cdot w_j)$$
$$= \epsilon \cdot (w_j - w_k) \cdot (\mathbb{E}_{\boldsymbol{v} \sim \Gamma} [v_x] - \mathbb{E}_{\boldsymbol{v} \sim \Gamma} [v_y]) \geq 0.$$

By repeating this type of update, the policy eventually becomes fully sorted, weakly increasing the utility with every step. Thus, the optimal policy must be sorted by $\mathbb{E}_{\boldsymbol{v} \sim \Gamma} [v_x]$. $\qquad\square$

**Lemma 3.1** *Consider two ranking policies $\pi_1$ and $\pi_2$ such that $\pi_1$ is $\phi_1$-fair and $\pi_2$ is $\phi_2$-fair. A policy that randomizes between $\pi_1$ and $\pi_2$ with probabilities $q$ and $1 - q$, respectively, is at least $(q\phi_1 + (1-q)\phi_2)$-fair and obtains expected utility $qU(\pi_1 \,|\, \Gamma) + (1-q)U(\pi_2 \,|\, \Gamma)$.*

*Proof.* Both the utility and fairness proofs are straightforward. The proof of fairness decomposes the probability of agent $i$ being in position $k$ under the mixing policy into the two constituent parts, then pulls terms through the sum. The proof of the utility bound uses Equation (2) and linearity of expectations. We now give details of the proofs.

We write $\pi_{\text{Mix}}$ for the policy that randomizes between $\pi_1$ and $\pi_2$ with probabilities $q$ and $1 - q$, respectively. Using Equation (2), we can express the utility of $\pi_{\text{Mix}}$ as

$$U(\pi_{\text{Mix}} \,|\, \Gamma) = \mathbb{E}_{\sigma \sim \pi_{\text{Mix}}, \boldsymbol{v} \sim \Gamma} [U(\sigma \,|\, \boldsymbol{v})]$$

$$= \mathbb{E}_{\boldsymbol{v} \sim \Gamma} \left[ \sum_\sigma \pi_{\text{Mix}}(\sigma) \cdot U(\sigma \,|\, \boldsymbol{v}) \right]$$

$$= \mathbb{E}_{\boldsymbol{v} \sim \Gamma} \left[ \sum_\sigma (q \cdot \pi_1(\sigma) + (1-q) \cdot \pi_2(\sigma)) \cdot U(\sigma \,|\, \boldsymbol{v}) \right]$$

$$= q \cdot \mathbb{E}_{\boldsymbol{v} \sim \Gamma} \left[ \sum_\sigma \pi_1(\sigma) \cdot U(\sigma \,|\, \boldsymbol{v}) \right] + (1-q) \cdot \mathbb{E}_{\boldsymbol{v} \sim \Gamma} \left[ \sum_\sigma \pi_2(\sigma) \cdot U(\sigma \,|\, \boldsymbol{v}) \right]$$

$$= qU(\pi_1 \,|\, \Gamma) + (1-q)U(\pi_2 \,|\, \Gamma).$$

Similarly, we prove that $\pi$ is at least $(q\phi_1 + (1-q)\phi_2)$-fair if $\pi_1$ and $\pi_2$ are $\phi_1$- and $\phi_2$-fair, respectively:

$$\sum_{k'=1}^k p_{x,k'}^{(\pi)} = \sum_{k'=1}^k q \cdot p_{x,k'}^{(\pi_1)} + (1-q) \cdot p_{x,k'}^{(\pi_2)}$$

$$= q \cdot \sum_{k'=1}^k p_{x,k'}^{(\pi_1)} + (1-q) \cdot \sum_{k'=1}^k p_{x,k'}^{(\pi_2)}$$

$$\geq q\phi_1 \cdot \mathbb{P}_{\boldsymbol{v} \sim \Gamma} \left[ \mathcal{M}_{x,k}^{(\boldsymbol{v})} \right] + (1-q)\phi_2 \cdot \mathbb{P}_{\boldsymbol{v} \sim \Gamma} \left[ \mathcal{M}_{x,k}^{(\boldsymbol{v})} \right]$$

$$= (q \cdot \phi_1 + (1-q) \cdot \phi_2) \cdot \mathbb{P}_{\boldsymbol{v} \sim \Gamma} \left[ \mathcal{M}_{x,k}^{(\boldsymbol{v})} \right],$$

where the inequality used that $\pi_1$ is $\phi_1$-fair and $\pi_2$ is $\phi_2$-fair. Hence, we have proved that $\pi$ is $(q \cdot \phi_1 + (1-q) \cdot \phi_2)$-fair under $\Gamma$. $\qquad\square$

**Lemma 3.2** *All 1-fair ranking policies have the same utility for the principal.*

*Proof.* Let $\pi$ be a 1-fair ranking policy. By Equation (1), $\pi$ must satisfy the following constraints:

$$\sum_{k'=1}^{k} p_{x,k'}^{(\pi)} \geq \mathbb{P}_{\boldsymbol{v} \sim \Gamma} \left[ \mathcal{M}_{x,k}^{(\boldsymbol{v})} \right] \qquad\qquad \text{for all } x \text{ and } k. \qquad (4)$$

Summing over all $x$ (for any fixed $k$), both the left-hand side and right-hand side sum to $k$; for the left-hand side, this is the expected number of agents placed in the top $k$ positions by $\pi$, while for the right-hand side, it is the expected number of agents among the top $k$ in merit. Because the weak inequality (4) holds for all $x$ and $k$, yet the sum over $x$ is equal, *each* inequality must hold with *equality*:

$$\sum_{k'=1}^{k} p_{x,k'}^{(\pi)} = \mathbb{P}_{\boldsymbol{v} \sim \Gamma} \left[ \mathcal{M}_{x,k}^{(\boldsymbol{v})} \right] \qquad\qquad \text{for all } x \text{ and } k.$$

This implies that

$$p_{x,k}^{(\pi)} = \mathbb{P}_{\boldsymbol{v} \sim \Gamma} \left[ \mathcal{M}_{x,k}^{(\boldsymbol{v})} \right] - \mathbb{P}_{\boldsymbol{v} \sim \Gamma} \left[ \mathcal{M}_{x,k-1}^{(\boldsymbol{v})} \right],$$

which is completely determined by $\Gamma$. Substituting these values of $p_{x,k}^{(\pi)}$ into the principal's utility, we see that it is independent of the specific 1-fair policy used. $\qquad\square$

**Proposition C.1.** Consider an algorithm that draws $m = \frac{(\kappa+1)\log(2n)}{2\epsilon^2}$ i.i.d. samples of the agents' joint merits from $\Gamma$, and then estimates each probability $\mathbb{P}_{\boldsymbol{v} \sim \Gamma} \left[ \mathcal{M}_{x,k}^{(\boldsymbol{v})} \right]$ by the empirical frequency with which $x$ was in position $k$ or higher. Then, with probability at least $1 - n^{-\kappa}$, all $\mathbb{P}_{\boldsymbol{v} \sim \Gamma} \left[ \mathcal{M}_{x,k}^{(\boldsymbol{v})} \right]$ are estimated with additive error at most $\pm\epsilon$.

*Proof.* Focus on one agent $x$, and write $q_k = \mathbb{P}_{\boldsymbol{v} \sim \Gamma} \left[ \mathcal{M}_{x,k}^{(\boldsymbol{v})} \right]$. Notice that the $q_k$ form the CDF of the rank of $x$. Let $Z_{k,j} = 1$ iff $x$ is among the top $k$ agents (by merit) in the $j^{\text{th}}$ of the $m$ samples. Then, $\mathbb{P}[Z_{k,j} = 1] = q_k$, and the estimate $Z_k = \frac{1}{m} \cdot \sum_j Z_{k,j}$ is the average of $m$ independent $\text{Bin}(q_k)$ random variables. By the DKW Inequality for the uniform convergence of the empirical CDF to the true CDF (Dvoretzky et al., 1956; Massart, 1990), we get that with probability at least $1 - 2\exp(-2m\epsilon^2) \geq 1 - n^{-(\kappa+1)}$, all of the estimates $Z_k$ are within $\pm\epsilon$ of the true values $\mathbb{P}_{\boldsymbol{v} \sim \Gamma} \left[ \mathcal{M}_{x,k}^{(\boldsymbol{v})} \right]$. A union bound over all $n$ agents now completes the proof. $\qquad\square$

While the estimates may be off by additive $\epsilon$ terms, it is fairly easy to compensate for such errors at a small loss in fairness and utility, as follows:

**Proposition C.2.** For each $x, k$, let $q_{x,k}$ be an empirical estimate of $\mathbb{P}_{\boldsymbol{v} \sim \Gamma} \left[ \mathcal{M}_{x,k}^{(\boldsymbol{v})} \right]$ such that $|q_{x,k} - \mathbb{P}_{\boldsymbol{v} \sim \Gamma} \left[ \mathcal{M}_{x,k}^{(\boldsymbol{v})} \right]| \leq \epsilon$ and $\sum_x q_{x,k} = k$ for all $k$. Consider the solution to the LP (3) with fairness parameter $\phi$, using[5] $q'_{x,k} = \frac{k(q_{x,k}+\epsilon)}{k+n\epsilon}$ in place of the (unknown) $\mathbb{P}_{\boldsymbol{v} \sim \Gamma} \left[ \mathcal{M}_{x,k}^{(\boldsymbol{v})} \right]$. Then, the resulting sampling distribution is at least $(\frac{\phi}{1+n\epsilon})$-fair, and guarantees the principal a utility within a factor $\frac{1}{1+n\epsilon}$ of the optimum $\phi$-fair solution.

*Proof.* First, notice that by the assumption that the $q_{x,k}$ were good approximations for $\mathbb{P}_{\boldsymbol{v} \sim \Gamma} \left[ \mathcal{M}_{x,k}^{(\boldsymbol{v})} \right]$, we can bound that $q'_{x,k} \geq \frac{k}{k+n\epsilon} \cdot \mathbb{P}_{\boldsymbol{v} \sim \Gamma} \left[ \mathcal{M}_{x,k}^{(\boldsymbol{v})} \right]$.

---

[5]Notice that the $q'_{x,k}$ in fact satisfy that $\sum_x q'_{x,k} = \frac{k}{k+n\epsilon} \sum_x (q_{x,k} + \epsilon) = \frac{k}{k+n\epsilon} \cdot (k + n\epsilon) = k$, so they can be used as input to the LP.

Because the LP's solution $(p_{x,k})_{x,k}$ is $\phi$-fair with respect to the $q'_{x,k}$, we get that

$$\sum_{k'=1}^{k} p_{x,k'} \;\geq\; \phi \cdot q'_{x,k} \;\geq\; \frac{k\phi}{k+n\epsilon} \cdot \mathbb{P}_{\boldsymbol{v} \sim \Gamma}\left[\mathcal{M}_{x,k}^{(\boldsymbol{v})}\right] \;\geq\; \frac{\phi}{1+n\epsilon} \cdot \mathbb{P}_{\boldsymbol{v} \sim \Gamma}\left[\mathcal{M}_{x,k}^{(\boldsymbol{v})}\right]$$

for all $x, k$; thus, the solution is $\left(\frac{\phi}{1+n\epsilon}\right)$-fair.

Next, we analyze the principal's utility. Let $(p_{x,k}^*)_{x,k}$ be a $\phi$-fair solution maximizing the principal's utility, and write $z_{x,k}^* = \sum_{k'=1}^{k} p_{x,k'}^*$ for the probability that agent $x$ is ranked among the top $k$ positions in the optimum solution. Now define $z'_{x,k} = \min(z_{x,k}^*, k - \phi \cdot \sum_{x' \neq x} q'_{x',k})$.

We will prove the following two facts: (1) The principal's utility under the probabilities $z'_{x,k}$ is not much smaller than under the original $z_{x,k}^*$, and (2) Every feasible solution $(p_{x,k})_{x,k}$ to the LP with fairness parameter $\phi$ and $q'_{x,k}$ satisfies $\sum_{k' \leq k} p_{x,k'} \geq z'_{x,k}$ for all $x, k$.

1. To show the first claim, we first use a standard way to rewrite the principal's objective in terms of the $z_{x,k}^*$ (or $z'_{x,k}$), using the definition $z_{x,0}^* := z'_{x,0} := 0$:

$$\sum_{x} \sum_{k=1}^{n} p_{x,k}^* \cdot \mathbb{E}_{\boldsymbol{v} \sim \Gamma}[v_x] \cdot w_k = \sum_{x} \mathbb{E}_{\boldsymbol{v} \sim \Gamma}[v_x] \cdot \sum_{k=1}^{n} (z_{x,k}^* - z_{x,k-1}^*) \cdot w_k$$

$$= \sum_{x} \mathbb{E}_{\boldsymbol{v} \sim \Gamma}[v_x] \cdot \left(\sum_{k=1}^{n} z_{x,k}^* \cdot w_k - \sum_{k=0}^{n-1} z_{x,k}^* \cdot w_{k+1}\right)$$

$$= \sum_{x} \mathbb{E}_{\boldsymbol{v} \sim \Gamma}[v_x] \cdot \left(w_n + \sum_{k=1}^{n-1} z_{x,k}^* \cdot (w_k - w_{k+1})\right). \quad (5)$$

Because $z'_{x,k} \leq z_{x,k+1}^*$ for all $x, k$, writing $p'_{x,k} := z'_{x,k} - z'_{x,k-1}$, we can also express the principal's utility under $(z'_{x,k})_{x,k}$ in the same way, simply replacing the terms $z_{x,k}^*$ with $z'_{x,k}$ in (5). Note that the $p'_{x,k}$ do not form a valid solution to the LP, because the "probabilities" do not necessarily sum up to 1 each across agents or across positions. However, we are only using this "solution" to help with our bounds, and feasibility is not required.

We can write the principal's loss in utility going from $z_{x,k}^*$ to $z'_{x,k}$ as follows:

$$\sum_{x} \mathbb{E}_{\boldsymbol{v} \sim \Gamma}[v_x] \cdot \left(w_n + \sum_{k=1}^{n-1} z_{x,k}^* \cdot (w_k - w_{k+1})\right)$$

$$- \sum_{x} \mathbb{E}_{\boldsymbol{v} \sim \Gamma}[v_x] \cdot \left(w_n + \sum_{k=1}^{n-1} z'_{x,k} \cdot (w_k - w_{k+1})\right)$$

$$= \sum_{x} \mathbb{E}_{\boldsymbol{v} \sim \Gamma}[v_x] \cdot \sum_{k=1}^{n-1} (z_{x,k}^* - z'_{x,k}) \cdot (w_k - w_{k+1})$$

$$= \sum_{k=1}^{n-1} (w_k - w_{k+1}) \cdot \sum_{x} \mathbb{E}_{\boldsymbol{v} \sim \Gamma}[v_x] \cdot (z_{x,k}^* - z'_{x,k}). \quad (6)$$

Notice that $w_k - w_{k+1} \geq 0$ for all $k$, and $\mathbb{E}_{\boldsymbol{v} \sim \Gamma}[v_x] \geq 0$ for all $x$. To upper-bound the loss in utility, we therefore can apply bounds for each of the terms $z_{x,k}^* - z'_{x,k}$. Focus on one particular pair $x, k$. Notice that the LP constraints (specifically, the third constraint and the first constraint) imply that

$$z_{x,k}^* \;=\; k - \sum_{x' \neq x} z_{x',k}^* \;\leq\; k - \phi \cdot \sum_{x' \neq x} \mathbb{P}_{\boldsymbol{v} \sim \Gamma}\left[\mathcal{M}_{x',k}^{(\boldsymbol{v})}\right] \;=\; k - \phi \cdot (k - \mathbb{P}_{\boldsymbol{v} \sim \Gamma}\left[\mathcal{M}_{x,k}^{(\boldsymbol{v})}\right]).$$

If $z'_{x,k} < z^*_{x,k}$, then

$$z'_{x,k} = k - \phi \cdot \sum_{x' \neq x} q'_{x',k} = k - \phi \cdot \sum_{x' \neq x} \frac{k(q_{x,k} + \epsilon)}{k + n\epsilon} = k - \frac{k\phi}{k + n\epsilon} \cdot (k - q_{x,k} + (n-1)\epsilon).$$

Therefore, the difference is at most

$$
\begin{aligned}
z^*_{x,k} - z'_{x,k} &\leq \frac{k\phi}{k + n\epsilon} \cdot (k - q_{x,k} + (n-1)\epsilon) - \phi \cdot \left(k - \mathbb{P}_{\boldsymbol{v} \sim \Gamma}\left[\mathcal{M}^{(\boldsymbol{v})}_{x,k}\right]\right) \\
&= \frac{\phi}{k + n\epsilon} \cdot \left((k^2 - kq_{x,k} + k(n-1)\epsilon) \right. \\
&\qquad\qquad \left. - (k^2 + kn\epsilon - (k + n\epsilon) \cdot \mathbb{P}_{\boldsymbol{v} \sim \Gamma}\left[\mathcal{M}^{(\boldsymbol{v})}_{x,k}\right])\right) \\
&\overset{(*)}{\leq} \frac{\phi}{k + n\epsilon} \cdot \left((-k(\mathbb{P}_{\boldsymbol{v} \sim \Gamma}\left[\mathcal{M}^{(\boldsymbol{v})}_{x,k}\right] - \epsilon) - k\epsilon) + (k + n\epsilon) \cdot \mathbb{P}_{\boldsymbol{v} \sim \Gamma}\left[\mathcal{M}^{(\boldsymbol{v})}_{x,k}\right])\right) \\
&= \frac{\phi n\epsilon}{k + n\epsilon} \cdot \mathbb{P}_{\boldsymbol{v} \sim \Gamma}\left[\mathcal{M}^{(\boldsymbol{v})}_{x,k}\right] \\
&\overset{(**)}{\leq} \frac{n\epsilon}{k + n\epsilon} \cdot z^*_{x,k}.
\end{aligned}
$$

Here, the line labeled (*) used that the $q_{x,k}$ approximate the true probabilities $\mathbb{P}_{\boldsymbol{v} \sim \Gamma}\left[\mathcal{M}^{(\boldsymbol{v})}_{x,k}\right]$ to within additive error at most $\epsilon$, and the line labeled (**) used that the $z^*_{x,k}$ formed a $\phi$-fair solution.[6]

We now substitute this bound (for each $k, x$) into (6), obtaining that the principal's loss in utility is at most

$$
\begin{aligned}
\sum_{k=1}^{n-1} (w_k - w_{k+1}) \cdot \sum_x \mathbb{E}_{\boldsymbol{v} \sim \Gamma}[v_x] \cdot \frac{n\epsilon}{k + n\epsilon} \cdot z^*_{x,k} \\
\leq \frac{n\epsilon}{1 + n\epsilon} \sum_{k=1}^{n-1} (w_k - w_{k+1}) \cdot \sum_x \mathbb{E}_{\boldsymbol{v} \sim \Gamma}[v_x] \cdot z^*_{x,k},
\end{aligned}
$$

which is exactly $\frac{n\epsilon}{1+n\epsilon}$ times the principal's utility under the solution $z^*_{x,k}$, i.e., the optimal utility. Thus, the utility obtained from using the approximate values is within at least a factor $1 - \frac{n\epsilon}{1+n\epsilon} = \frac{1}{1+n\epsilon}$ of optimal.

2. Next, we show that every feasible solution $(p_{x,k})_{x,k}$ to the LP with fairness parameter $\phi$ and $q'_{x,k}$ satisfies $\sum_{k' \leq k} p_{x,k'} \geq z'_{x,k}$ for all $x, k$. In fact, we show that $\sum_{k' \leq k} p_{x,k'} \geq k - \phi \cdot \sum_{x' \neq x} q'_{x',k}$, which in turn is at least $z'_{x,k}$ by definition of $z'_{x,k}$.

To see this, note that for any feasible solution and for all $x, k$, the fairness constraint implies that $\sum_{k'=1}^{k} p_{x,k'} \geq \phi \cdot q'_{x,k}$ and furthermore, $\sum_x p_{x,k'} = 1$ for all $k'$. Therefore, for any fixed $x, k$,

$$k = \sum_{x'} \sum_{k'=1}^{k} p_{x',k'} = \sum_{k'=1}^{k} p_{x,k'} + \sum_{x' \neq x} \sum_{k'=1}^{k} p_{x',k'} \geq \sum_{k'=1}^{k} p_{x,k'} + \sum_{x' \neq x} \phi \cdot q'_{x',k}.$$

Rearranging this inequality gives us the claimed bound.

Now consider the optimal solution $p_{x,k}$ (maximizing the principal's utility) with fairness parameter $\phi$ and estimated probabilities $q'_{x,k}$. For each $x, k$, define $z_{x,k} = \sum_{k'=1}^{k} p_{x,k'}$. Then, $z_{x,k} \geq z'_{x,k}$ for all $x, k$, and the utility under $(p_{x,k})_{x,k}$ is given by (5) (with $z_{x,k}$ in place of $z^*_{x,k}$). In particular, it is at least as large as under $(z'_{x,k})_{x,k}$, and thus within a factor of $\frac{1}{1+n\epsilon}$ of the optimum. $\qquad\square$

---

[6]For $\phi = 0$, the calculations do not apply, but in that case, the algorithm can completely ignore the estimated probabilities, and will obtain the optimum solution.

By Proposition C.2, if the principal wants to approximate fairness and utility to within a factor $1 - \epsilon$, it suffices to approximate the $\mathbb{P}_{\boldsymbol{v} \sim \Gamma}\left[\mathcal{M}_{x,k}^{(\boldsymbol{v})}\right]$ to within an additive error of at most $\frac{\epsilon}{n(1-\epsilon)}$. In turn, by Proposition C.1, it is sufficient to draw $O(\frac{\kappa n^2 \log n}{2\epsilon^2})$ samples from $\Gamma$ to achieve this approximation with probability at least $1 - n^{-\kappa}$; in particular, the number is polynomial in $n$ and $1/\epsilon$.

# D    Example for Suboptimality of $\pi^{\mathrm{Mix},\phi}$

Here, we give an example showing that the policy $\pi^{\mathrm{Mix},\phi}$ may not yield optimal utility for the principal among $\phi$-fair policies. The example illustrates the types of tradeoffs to be considered for approximately fair solutions, and motivates the LP-based efficient algorithm in § 3.2.

**Example 2.** Consider $n = 3$ agents, namely $a, b$, and $c$. Under $\Gamma$, their merits $v_a = 1$, $v_b \sim$ Bernoulli($1/2$), and $v_c \sim$ Bernoulli($1/2$) are drawn independently.[7] The position weights are $w_1 = 1$, $w_2 = 1$, and $w_3 = 0$.

Now, since $w_1 = w_2 = 1$ and agents $b$ and $c$ are i.i.d., any policy that always places agent $a$ in positions 1 or 2 is optimal. In particular, this is true for the policy $\pi^*$ which chooses uniformly at random from among $\sigma_1^* = \langle a, b, c\rangle$, $\sigma_2^* = \langle a, c, b\rangle$, $\sigma_3^* = \langle b, a, c\rangle$, and $\sigma_4^* = \langle c, a, b\rangle$.

For the specific distribution $\Gamma$, assuming uniformly random tie breaking, we can calculate the probabilities $\mathbb{P}_{\boldsymbol{v} \sim \Gamma}\left[\mathcal{M}_{x,k}^{(\boldsymbol{v})}\right]$ in closed form:

$$\left(\mathbb{P}_{\boldsymbol{v} \sim \Gamma}\left[\mathcal{M}_{x,k}^{(\boldsymbol{v})}\right]\right)_{x,k} = 1/24 \cdot \begin{pmatrix} 14 & 22 & 24 \\ 5 & 13 & 24 \\ 5 & 13 & 24 \end{pmatrix}.$$

The probability of $a$, $b$, $c$ being *placed* in the top $k$ positions by $\pi^*$ can be calculated as follows:

$$\mathcal{P}^{(\pi^*)} = 1/24 \cdot \begin{pmatrix} 12 & 24 & 24 \\ 6 & 12 & 24 \\ 6 & 12 & 24 \end{pmatrix}.$$

In particular, this implies that $\pi^*$ is $\phi$-fair for every $\phi \leq 12/14 = 6/7$. This bound can be pushed up by slightly increasing the probability of ranking agent $a$ at position 1 (hence increasing fairness to agent $a$ in position 1 at the expense of agents $b$ and $c$ in positions 1–2). Figure 3 shows the principal's optimal utility for different fairness parameters $\phi$, derived from the LP (3). This optimal utility is contrasted with the utility of $\pi^{\mathrm{Mix},\phi}$, which is the convex combination of the utilities of $\pi^*$ and $\pi^{\mathrm{TS}}$, by Lemma 3.1.

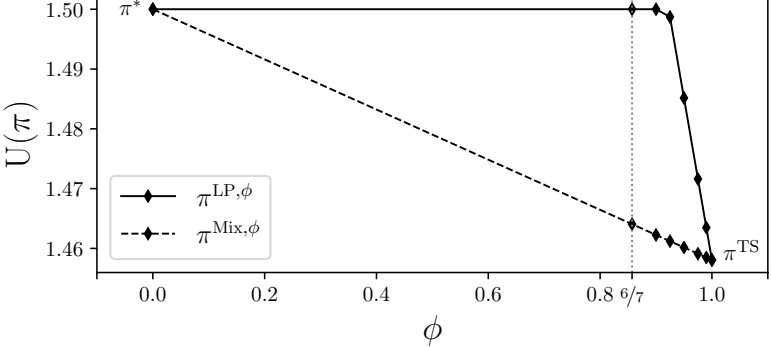

Figure 3: Utility of $\pi^{\mathrm{Mix},\phi}$ and $\pi^{\mathrm{LP},\phi}$ for Example 2 as one varies $\phi$.

---

[7]Technically, this distribution violates the assumption of non-identical merit of agents under $\Gamma$. This is easily remedied by adding — say — i.i.d. $\mathcal{N}(0, \epsilon)$ Gaussian noise to all $v_i$, with very small $\epsilon$. We omit this detail since it is immaterial and would unnecessarily overload notation.

# E  Details on the MovieLens Dataset Experiment

The MovieLens-100k dataset contains 100,000 ratings, by 600 users, on 9,000 movies belonging to 18 genres (Harper and Konstan, 2015). In our setup, for each user, the principal is a recommender system that has to generate a ranking of movies for one of the genres $g$ (e.g., Horror, Romance, Comedy, etc.), according to a notion of *merit* of the movies we define as follows.

## E.1  Experimental Setup

We assume that each rating of a movie $m \in \mathcal{S}_g$ is drawn from a multinomial distribution over $\{1, 2, 3, 4, 5\}$ with (unknown) parameters $\boldsymbol{\theta}_m = (\theta_{m,1}, \ldots, \theta_{m,5})$.

**Prior**: These parameters themselves follow a Dirichlet prior $\boldsymbol{\theta}_m \sim \text{Dir}(\boldsymbol{\alpha})$ with known parameters $\boldsymbol{\alpha} = (\alpha_1, \alpha_2, \alpha_3, \alpha_4, \alpha_5)$. We assume that the parameters of the Dirichlet prior are of the form $\alpha_r = s \cdot p_r$ where $s$ is a scaling factor and $p_r = \mathbb{P}[\text{Rating} = r \mid \mathcal{D}]$ denotes the marginal probability of observing the rating $r$ in the full MovieLens dataset.

The scaling factor $s$ determines the weight of the prior compared to the observed data, since it acts as a pseudo-count in $\boldsymbol{\alpha}'$ below. For the sake of simplicity, we use $s = 1$ in the following for all movies and genres.

**Posterior**: Since the Dirichlet distribution is the conjugate prior of the multinomial distribution, the posterior distribution based on the ratings observed in the dataset $\mathcal{D}$ is also a Dirichlet distribution, but with parameters $\boldsymbol{\alpha}' = (\boldsymbol{\alpha} + \boldsymbol{N}_m) = (\alpha_1 + N_{m,1}, \ldots, \alpha_5 + N_{m,5})$ where $N_{m,r}$ is the number of ratings of $r$ for the movie $m$ in the dataset $\mathcal{D}$.

## E.2  Expected Merit

The optimal ranking policy $\pi^*$ sorts the movies (for the particular query) by decreasing expected merit, which is the expected average rating $\overline{v}_m$ under the posterior Dirichlet distribution, and can be computed in closed form as follows:

$$\overline{v}_m \triangleq \mathbb{E}_{\boldsymbol{\theta} \sim \mathbb{P}[\boldsymbol{\theta}_m \mid \mathcal{D}]}[v_m(\boldsymbol{\theta})] = \sum_{r=1}^{5} r \cdot \frac{\alpha_r + N_{m,r}}{\sum_{r'} \alpha_{r'} + N_{m,r'}}. \tag{7}$$

## E.3  Ranking Distribution Visualization

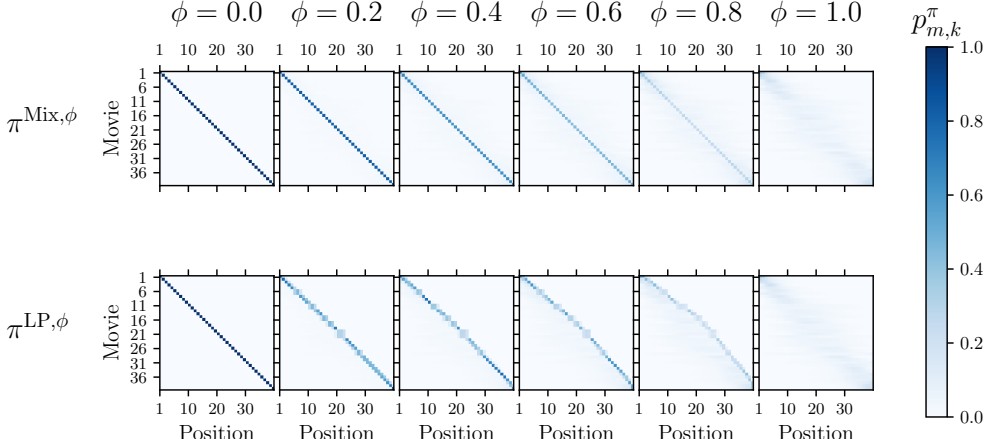

Figure 4: Comparison of marginal rank distribution matrices for $\pi^{\text{Mix},\phi}$ and $\pi^{\text{LP},\phi}$ on "Comedy" movies.

To provide intuition about the difference between the solutions of the LP and OPT/TS-Mixing for different values of $\phi$, we visualize $\pi^{\text{Mix},\phi}$ and $\pi^{\text{LP},\phi}$ in Figure 4. The plotted matrices are the marginal

rank distributions: $p_{m,k}$ represents the probability that movie $m$ is ranked at position $k$. Note that the distribution at $\phi = 0$ and $\phi = 1$ is identical for the two methods, as shown in our lemmas. The key distinction between the rank distributions for $\phi \in (0,1)$ is that $\pi^{\mathrm{LP},\phi}$ finds non-linear structure for intermediate values of $\phi$, while $\pi^{\mathrm{Mix},\phi}$ merely interpolates linearly between the solutions for $\phi = 0$ and $\phi = 1$.

# F    Details on the Real-World Experiment

As described in § 5, we designed a real-world experiment through a paper recommendation system where the users were the participants at the 2020 ACM SIGKDD Conference on Knowledge Discovery and Data Mining. Signup and usage of the system was voluntary.

Each user was recommended a personalized ranking of the papers published at the conference. This ranking was produced either by $\sigma^*$ or by $\pi^{\mathrm{TS}}$, and the assignment of users to treatment ($\pi^{\mathrm{TS}}$) or control ($\sigma^*$) was randomized.

## F.1    Users of the Paper Recommendation System

A total of 923 users signed up for the system ahead of the conference (and were randomized into treatment and control groups). Out of these 923 users, 462 did not use the system after all. Of the users that logged in at least once, 213 users were in $\mathcal{U}_{\pi^*}$, and 248 users were in $\mathcal{U}_{\pi^{\mathrm{TS}}}$. Note that this difference is not caused by the treatment assignment, since users had no information about their assignment before entering the system. Users could either navigate through their recommendations by clicking next or previous buttons on their recommendation page, or had other options to engage with each paper such as reading the abstract, reading the PDF, adding the paper to their calendar, and adding a bookmark to the paper.

## F.2    Modeling the Merit Distribution

The merit of a paper for a particular user is based on a relevance score $\mu_{u,i}$ that relates features of the user (e.g., bag-of-words representation of recent publications, co-authorship) to features of each conference paper (e.g., bag-of-words representation of paper, citations). Most prominently, the relevance score $\mu_{u,i}$ contains the TFIDF-weighted cosine similarity between the bag-of-words representations.

We model the uncertainty in $\mu_{u,i}$ with regard to the true relevance as follows. First, we observe that all papers were accepted to the conference and thus must have been deemed relevant to at least some fraction of the audience by the peer reviewers. This implies that papers with uniformly low $\mu_{u,i}$ across all/most participants are not irrelevant; we merely have high uncertainty as to which participants the papers are relevant to. For example, papers introducing new research directions or bringing in novel techniques may have uniformly low scores $\mu_{u,i}$ under the bag-of-words model that is less certain about who wants to read these papers compared to papers in established areas. To formalize uncertainty, we make the assumption that a paper's relevance to a user follows a normal distribution centered at $\mu_{u,i}$, and with standard deviation equal to $\delta_i$ (dependent only on the paper, not the user) such that $\max_u \mu_{u,i} + \gamma \cdot \delta_i = 1 + \epsilon$. (For our experiments, we chose $\epsilon = 0.1$ and $\gamma = 2$.) This choice of $\delta_i$ ensures that there exists at least one user $u$ such that the (sampled) relevance score $\hat{\mu}_{u,i}$ is greater than 1 with some significant probability; more specifically, we ensure that the probability of having relevance $1 + \epsilon$ is at least as large as that of exceeding the mean by two standard deviations. Furthermore, $\epsilon > 0$ ensures that all papers have a non-deterministic relevance distribution, even papers with $\max_u \mu_{u,i} = 1$.