# OpenReview forum: "Fairness in Ranking under Uncertainty"
_NeurIPS.cc/2021/Conference — NeurIPS 2021 Poster_

### Official Review · Reviewer_7pR9 · 2021-07-14

**Rating:** 6
**Confidence:** 5

**Summary:**

The paper proposes a new definition of fairness in rankings. An LP-based algorithm is given for finding best (max utility) ranking for a given fairness level. Two empirical evaluations of the proposed frameworks are given 1) based on ranking movies from a public dataset and 2) an experiment was run during a large virtual conference in which participants were given paper recommendations according to a ranking.



**Limitations And Societal Impact:**

-

**Main Review:**

The main assumption (or belief) behind this paper is that there is inherent uncertainty in the ranked items' merits which are observed by the ranker. This uncertainty is modelled by a probability distribution over merits that is given as input to the ranker. Also the solution is then not a single permutation over items but rather a whole distribution over them. A (candidate solution) distribution over rankings is said to be \phi-fair if for every item x and every position k in the ranking, the probability of x being ranked in top-k is at least \phi times the probability of x being top-k according to the given distribution of merits.

What is important to point out is that for the standard setting of a ranking problem, where *a single ranking must be produced* based on some kind of data, this framework does not seem applicable -- the main point of having a probabilistic solution is that one samples from it repeatedly, multiple times, and not just once. It is mentioned in the paragraph 180-185 that because of this issue the applicability of this approach is limited. In the reviewer's opinion this is a rather strong limitation, and the paper makes the impression as if the "framework" was very general -- it would be certainly beneficial to state upfront what is the setting the framework is expected to be applied to, because it is certainly not the most common one.

The mathematical formulation of the fairness notion proposed by the authors is quite nice and ellegant. This ellegance also translates to a very natural LP formulation of the problem that the authors give as a solution method. However, several natural questions come to mind:
1) The knowledge of Prob[M^{(v)}_{x, k}] is a very strong assumption, it is not at all convincing that this can be obtained -- and even if it can by MC, then the error can significantly impact the optimal solution.
2) There is no discussion of other, perhaps combinatorial approaches to solve this problem. Is it obviosly hard? Is LP somehow necessary?
3) Even if LP is necessary, and no greedy approach works, then giving a practical algorithm for ranking that involves solving an LP does not sound right. It would be great if the authors discussed some possible practical, efficient algorithmic approaches (or at least heuristics) for solving this linear program that could allow avoiding a general LP-solver.


When it comes to the experiments, first of all the reviewer is impressed by the Real World Experiment on paper recommendation (during a virtual conference). This is a great piece of work which deserves appreciation as it certainly required a lot of effort to implement. That being said, even though this single part of the paper makes a great, positive impression, the reviewer's overall feedback on the experiments, or rather the whole proposed "framework" is not quite positive. The main objection is that the premise of uncertainty and probabilistic modelling of the problem feels rather artificial and does not seem to arise naturally in real-world examples. This is unfortunately confirmed by the design forced in both experiments: in each of them, some kind of artificial randomness must be injected just so that the assumption of "uncertainty" is met. If this assumption was correct and natural, why would there ever be need to inject artificial randomness into the data?


To conclude, the paper is well written and the considered idea for a framework for ranking seems reasonable from the mathematical perspective. Also the obtained algorithm is natural, and the performed experiments are strong parts of the paper. However, the reviewer is of the opinion that the main assumption made in this paper -- that uncertainty is inherent in ranking problems -- in the way it is modelled in this framework, is doubtful, and somehow even the experiments reinforce this doubt (injecting artificial randomness) instead of proving the opposite.


Minor comments:

1. typo in abstract, line 17 'compute,rankings'
2. there are multiple papers that are referenced via arxiv, but have been published in conferences in the meantime

**Time Spent Reviewing:**

6

---

> ### Author Response · Authors · 2021-08-06
> **Responses to key issues raised in your review**
>
> Thank you for the very detailed and constructive review! You obviously spent significant effort understanding our work in detail, and you raise some important issues worthy of discussion.
> Several issues closely related to the ones you raise are also addressed in our response to Reviewer gSPf, which we encourage you to read (and will not duplicate here).
>
> 1. Most fundamentally, you question whether uncertainty in merit is indeed a real-world concern. We believe that the presence of uncertainty should be hard to dispute. Not all recommendations on e-commerce or other recommendation sites work out, not all hires or college admissions work out, and many good opportunities are missed. Most principals would make many different decisions in retrospect.
> More specifically, you claim that our experiments need to “artificially inject” randomness. For the MovieLens experiment, this is simply not true. The randomness here arises very naturally from distributions of star ratings in the data set, which we adjust only to capture that movies with few ratings have higher variance in merit.
> For the field experiment at a conference, our method is indeed somewhat convoluted. But this is not due to a weakness in the framework, but rather, a weakness in the data set. Data relating conference attendees with papers were very limited, and text-based similarity measures of papers are very crude. In principle, with huge effort (e.g., manual labeling by MTurkers), one could have obtained much better data, which would have much more explicitly exhibited distributions (e.g., what fraction of manual labels claimed that a paper would be relevant to particular attendees based on reading the paper and the attendee’s website). However, such an effort was beyond the scope of this project, in particular given that this experiment was supposed to serve mostly as a “proof of concept”.
>
> 2. Like Reviewer gSPf, you raise the issue of applicability of the approach in high-stakes single-shot settings, which we discuss in Lines 180-185. While we believe that there are serious practical concerns to address before deploying any randomized approach in such settings, we would not go so far as to say that the approach does not apply at all. As discussed in our response to gSPf, there will be situations where a principal wants to be fair choosing one out of two essentially identical agents. This problem of (approximate) tie-breaking is ubiquitous, for example, in college admissions at top universities, where qualified applicants greatly outnumber available slots. If we have two college applicants who are indistinguishable for practical purposes, but only one position, what is a ‘fair’ thing to do? We believe that even here, a coin flip may be considered the fairest approach (as opposed to breaking ties alphabetically by last name, or any other deterministic criterion). This situation must eventually be addressed, and while we see concerns with an implementation using randomization, we still believe that our fairness framework can give very useful guidance.
> As we elaborate in our response to gSPf, besides practical issues (“did the principal really randomize?”), we believe that a source of discomfort with randomization in single-shot settings is ex ante vs. ex post fairness. Before the coin is flipped, each agent has the same chance of being treated preferentially, so the coin flip is ex ante fair. But after the coin flip, one agent lost, so ex post, an equally deserving agent was treated worse. By contrast, in settings with repetition, fairness will be (approximately) achieved even ex post, as each agent will be treated preferentially in (approximately) the correct fraction of cases. We believe that this may be at the heart of your concern, expressed as the statement that the point of having a distribution is to sample from it multiple times. As we discussed above, achieving ex post fairness in a single-shot setting is really impossible, so implementing an approach that at least achieves ex ante fairness appears like a useful alternative, and this is indeed achieved by randomization.
>
> 3. You raise the issue of inefficiency of solving LPs. This is a valid concern, and we have several responses. First, for several hundred agents, LPs would be just fine in practice. Second, experimentally, the algorithm that mixes between TS and OPT is a good (and extremely efficient) alternative. Proving that it satisfies provable approximation guarantees (beyond the trivial ones one can obtain from invoking the optimum with a certain probability) is an interesting direction for future work. Third, as you suggest, if necessary, one could investigate faster alternatives for solving the LP, e.g., with greedy algorithms or sophisticated techniques that the theory community has analyzed for packing/covering LPs. Perhaps the LP can be solved optimally (or with provable approximation guarantees) using such approaches. We did not focus on the algorithm’s efficiency in our submission because we wanted to clearly focus on the conceptual contribution, and were using the algorithms to show (a) that computation is indeed possible for reasonable problem sizes, and (b) to run our proof-of-concept experiments. Future work by us or others could well focus on more efficient algorithms if the framework is more widely accepted.
>
> 4. You raise the issue of knowing the probabilities of M^(v)_{x,k}, which we discuss in Lines 243-249. While this is indeed an issue, within a reasonable (polynomial in 1/\epsilon) number of samples, an algorithm can estimate \epsilon-accurate probabilities. By making \epsilon=0.00001, the loss in fairness or utility would indeed be very small, and the sampling overhead quite small - each sample only involves drawing from a known distribution and sorting. So while this is an issue that does need to be considered, we do not believe that it is a major impediment.
>
> 5. We will of course update the arXiv citations, and have already done so in our own bibliography between the deadline and now.

---

> > ### Comment · Reviewer_7pR9 · 2021-08-21
> > **Thank you for response.**
> >
> > Thank you for a thorough and informative response. Both in this response as well as for reviewer gSPf you included some important points that I believe can make the paper better and more convincing. That being said, I'm afraid that I cannot increase my score as I still believe in the correctness of my main objection. Other than that, I like this paper a lot and I'm genuinely rooting for the authors, as they put a lot of effort in writing it, performing the experiments and even now: in submitting in-depth responses.

---

> > > ### Author Response · Authors · 2021-08-28
> > > **Thank you for the follow-up conversation**
> > >
> > > Thank you for your response, and for rooting for our paper!
> > >
> > > We are honestly still a little puzzled about your objection to the central premise of the paper that given observable features, there is uncertainty in merit. We feel that perhaps, we are misunderstanding your specific objection.
> > >
> > > To us, it seems completely self-evident that given observables (e.g. the curriculum vitae of a job candidate), there is always residual uncertainty about how the candidate will *actually* perform (e.g. at the 5-year evaluation); and to our community (CS/ML/statistics), it is most natural to model this uncertainty in a Bayesian way.
> > >
> > > But even if we disagree on the premise that uncertainty exists (and that some form of lottery can be useful in coming to a fair ranking), we do all agree that the paper is rigorous, novel, and well executed. So, at the very least, there seems to be common ground that our paper makes an interesting contribution by rigorously defining this issue, and that it will spark an interesting debate.

---

> > > > ### Comment · Reviewer_7pR9 · 2021-09-05
> > > > **Reading the paper again**
> > > >
> > > > I have read the paper again, this time spending more time in the experiments section. On the second look, I admit that I was too harsh (due to my misunderstanding of some aspects of Sections 5 and 6) in my initial evaluation. I'll increase my score by (1.5) points, rounding up -- thus it now stands at 6.
> > > >
> > > > My current impressions are:
> > > > -- I disliked the movie ranking example (now I dislike it less, but still...) as an example of practical application of your framework. This is partially because "fairness" in the context of movie recommendation does not sound like the most natural direction (the motivation for studying fairness was fighting discrimination etc.). Anyway, what still bothers me is that you need to make this, perhaps standard, but somehow arbitrary decision to use the Dirichlet prior -- this has impact on the conclusions you get in this experiment. I'm aware that this is how Bayesian models work, but I still, don't consider it completely natural...
> > > > -- The real-world experiment I liked quite a bit, and now, after clearing out some doubts, I like it even more.

---

> > > > > ### Author Response · Authors · 2021-09-06
> > > > > **Thank you for the constructive discussion**
> > > > >
> > > > > Thank you again for the thoughtful continuation of the discussion! We agree that the MovieLens dataset is best seen as a stand-in for more consequential problem domains. Complementing the affordances of this dataset was a key motivation for also going through the effort of building and evaluating the method in a real-world system. Regarding the Dirichlet prior, we wanted to stick to standard methodology as much as possible to avoid any impression of idiosyncratic choices in our proof-of-concept, but we agree that a practitioner may want to choose priors that are more tailored to the domain.

---

### Official Review · Reviewer_gSPf · 2021-07-16

**Rating:** 6
**Confidence:** 4

**Summary:**

The paper considers ranking given access to posterior distributions on true scores. They propose a notion of fairness based on these posterior distributions, essentially requiring that probability of inclusion among the top $k$ scale with the probability of being ranked in the top $k$ according to the posterior distributions. These posterior distributions are presumed to be fair, in the sense that an algorithm which is fair (according to this definition) need only be fair with respect to the posterior distributions.
They show that a convex combination of Thompson Sampling and ranking by means is approximately fair but can be suboptimal in terms of utility. They give an optimal algorithm (subject to their fairness constraint) by formulating the problem as a linear program. Finally, they run an experiment on a movie recommendation dataset and implement their algorithm in a real-world paper recommendation setting, comparing their two algorithms and measuring cost of fairness.


**Limitations And Societal Impact:**

As noted in the paper, bias mitigation—if it were to happen—would take place in choosing the posterior distributions, not in choosing the ranking. Some discussion on how one might mitigate a bias by choosing an appropriate posterior distribution would be helpful.



**Main Review:**

The paper is well-written and clear, presenting simple algorithms that can be easily implemented in practice. The biggest contribution is the notion of fairness, which requires that ordinal probabilities be bounded based on posterior distributions. This is a novel notion of fairness in ranking to my knowledge. The biggest drawback of the paper is that the technical (theory) contribution is somewhat weak.

Some concerns that arise due to the definition of fairness:
- A notion of fairness based purely on posterior distributions might enhance inequities in the society, simply due to the fact that the initial observations for certain agents might be luckier than for other agents.
- How are the posterior distributions computed? How might these distributions account for systemic biases? I wonder if the paper is simply by-passing the key issue in evaluation metrics by considering posterior distributions.
- Randomness -- probabilistic definition of ranking -- why does a probabilistic definition even make sense, when providing a single ranking for agents, for applications such as admissions, hiring, etc?
- A lot of references on rankings seem to be missing -- for e.g., Kleinberg and Raghavan (subset selection under bias), Bower et. al (individually fair rankings), Gupta and Salem (selection under poset model), Celis and Mehrotra (subset selection with noisy protected attributes), Celis Mehrotra Vishnoi (ranking under implicit bias), etc.

Minor comment: The discussion of error decay in lines 244-249 should be written formally as a proposition.


**Time Spent Reviewing:**

1 (by me), 3 (by my student)

---

> ### Author Response · Authors · 2021-08-06
> **Response to important issues raised in your review**
>
> Thank you for the detailed and thoughtful review! You raise two key points, also raised by Reviewer 7pR9, which we will respond to in detail, because they are so central to our approach.
>
> 1. How are posteriors computed, how could they account for systemic biases (or avoid exacerbating biases or unlucky initial outcomes), and aren’t we just side-stepping important questions here?
> We have a brief discussion of this issue in Lines 163-171, but given its central nature, perhaps, we should have given it more space. We will expand here (and in a final version).
> As you noticed, the task of estimating the posterior is indeed complex, and fraught with the potential for bias. However, any algorithmic solution to a problem, and in particular any that attempts to achieve fairness, shares this concern. Somewhere, based on observable features, a decision is made, and some agents receive a better outcome than others. The algorithm’s decisions, be they based on posterior merit distributions, metrics on observable features, or other mathematical approaches, lead to some outcomes - therefore, in a sense, the algorithm encodes normative judgments by its designers (and principals who set parameters or train it). The main question is then how these judgments are encoded, how they can be articulated, and whether they could be audited.
> A key feature of our approach lies in decoupling the definition of fairness from the problem of modeling the data, so that each can be discussed, validated and audited separately. This abstraction boundary with merit as its key concept adds conceptual clarity about how the normative choice of the fairness criterion differs from the choices regarding statistical modeling and estimation, and allows a practitioner to disentangle the two. A key question that must be answered before fairness can be truly considered is “What do the observable features tell us about a candidate’s merit?”, and this is precisely encoded in the posterior merit distribution.
> We believe that articulating a choice of the posterior (or a method - such as ML-based techniques - for arriving at a posterior) will be much more natural for domain experts than dealing with the intricate definitions inside algorithms. For example, if a college admissions officer believes that SAT scores of affluent applicants may likely be inflated due to access to tutors, she can explicitly encode this in a merit distribution - in turn, other officers or agencies can question this decision, or the specific probabilities.
> As computer scientists, we can help provide statistical tools to infer merit from data, but we do not believe that we are in a position to make specific prescriptions about the approaches that should be used - if nothing else, these will differ significantly between domains. For our experiments, we chose what we believe to be a plausible approach for inferring posteriors in order to obtain a proof of concept.
> But again, the main point is that we believe that all questions regarding the modeling of merit - including deliberate bias mitigation - should be pushed into the question of obtaining a suitable posterior, and that this question must be addressed in close collaboration with domain experts if fairness is to be achieved in real-world settings.
>
> 2. What is even the meaning of randomization in a single-shot setting such as hiring or college admissions?
> We also have a brief discussion of this in Lines 180-185. At a high level, the question raises a very fundamental philosophical question, perhaps best distilled as follows: “If we have two applicants who are indistinguishable for practical purposes, but only one position, what is a ‘fair’ thing to do?” We believe that even here, a coin flip may be considered the fairest approach. And especially in college admissions, where qualified applicants greatly outnumber available slots at most top universities, the problem of breaking ties between applicants (or dealing with uncertainty about merit) is ubiquitous. There may be significant practical concerns in high-stakes single-shot scenarios regarding randomization (see our discussion), but giving all applicants a “fair chance” still feels more fair than any deterministic approach. One way to precisely articulate why randomization “feels” more fair in repeated settings is the following: a single coin flip is fair ex ante: each agent has the same chance of winning. However, it is not fair ex post: one agent was treated worse after the coin was flipped. In a repeated setting, fairness is also achieved ex post: each agent obtained (approximately) the expected share of preferential treatment they should receive.
>
> 3. References we failed to cite.
> We will gladly expand our discussion of related work (which already runs to over 60 papers as is). Obviously, we made a significant effort to broadly cite related work, but we will further expand the radius of our related work discussion in the final version.

---

> > ### Comment · Reviewer_gSPf · 2021-08-31
> > **Keeping my score unchanged.**
> >
> > I appreciate the authors thoughtful and detailed response to some of the concerns raised in my review. I would however like to keep my score as is.

---

### Official Review · Reviewer_xirY · 2021-07-17

**Rating:** 7
**Confidence:** 3

**Summary:**

This paper studies the problem of fair ranking under a new notion of fairness. The main point of the paper is that uncertainty is a primary reason for the lack of fairness. In defining this new notion of fairness, authors base their arguments on the axiom that if $A$ has merit greater than or equal to $B$ with probability at least $p$, then a fair policy should treat $A$ at least as well as $B$ with probability at least $p$. Based on this, they say a ranking policy is fair if the probability that it ranks an agent in the first $k$ positions is at least the probability that it actually is among the top $k$ agents. They similarly say an algorithm is $\phi$-fair if the probability that it ranks an agent in the first $k$ positions is at least a $\phi$ fraction of the probability that it actually is among the top $k$ agents.
Authors then try to design algorithms for fair ranking. They first propose two very simple algorithms which achieve fair and approximately fair ranking without considering its impacts on the utility of the principal. They then show how to compute rankings that optimally trade-off approximate fairness against utility to the principal. Finally, they conduct empirical studies to evaluate the impact of their approach on a paper recommendation system that they built and fielded at a large conference.


**Limitations And Societal Impact:**

I don't believe that there is any negative social impact.

**Main Review:**

I find this new notion of fairness very interesting. The authors do a good job of explaining why thinking about fairness in this way is useful and how uncertainty plays a significant role in getting unfair policies. The main technical contribution of the paper is designing ranking policies which are quite trivial (So, the paper is not very technically strong). However, using empirical studies, the authors show that these simple algorithms give good results. In conclusion, I believe that the paper meets the bar for a NeurIPS publication mostly due to its novelty (studying this new notion of fairness).

A minor comment for the authors: There is a typo in the abstract “compute,rankings”.

**Time Spent Reviewing:**

3 hours

---

> ### Author Response · Authors · 2021-08-06
> **Response to your review**
>
> Thank you for your careful and considerate review! We agree with your assessment that the main contribution is conceptual: outlining the need to consider uncertainty and defining a notion of fairness. The algorithms are indeed simple, and serve as a “proof of concept”: one can achieve something meaningful within this framework. Of course, we would also like to stress that despite its triviality, the algorithm that mixes between the optimal and fair policies seems to perform well, and it is of course very efficient, and could thus be an option in practice.

---

### Official Review · Reviewer_4iKN · 2021-07-17

**Rating:** 7
**Confidence:** 2

**Summary:**

This paper studies the fairness issue in algorithmic decision-making caused by uncertainty. Since the actual merits are hard to observe, the authors propose a new notion to capture the uncertainty, named approximate fairness in ranking. Based on the notation, they propose different algorithms to achieve approximately fair ranking distributions. In addition to the theoretical analysis, extensive empirical studies are conducted to evaluate their approach.

**Limitations And Societal Impact:**

1. It would be better to show the running time of the proposed algorithm in the experimental section.
2. Comparisons with other state-of-the-art algorithms are missing, thus readers cannot see whether the proposed algorithms can outperform the related work.

**Main Review:**

This paper is well-written and studies an interesting problem, which is well explained in the introduction. To some extent, it will provide a new perspective for the related research communities to explore the fairness issue under uncertainty. This paper is technically sound for me and the empirical studies are versatile which including data-driven simulation and real implementation.

**Time Spent Reviewing:**

6

---

> ### Author Response · Authors · 2021-08-06
> **Response to your comments/suggestions**
>
> Thank you for the positive review and suggestions!
> In response to your two suggestions:
> 1. We will be happy to include running times in the final version. We left them out of the submission because one algorithm consists just of sampling+sorting (so its running time is obviously O(n log n)), while the other is based just on solving an LP, which also has well understood running time. Furthermore, our main contribution is the new framework (of considering uncertainty as a first-order concern for fairness) - we did not want to give too much emphasis to the specific algorithms.
> 2. We did not compare to other algorithms, as we are not aware of other approaches/frameworks that explicitly model and address uncertainty in its implications for fairness. And, clearly, any deterministic algorithm or any algorithm that uses point estimates would be considered very unfair in our setting.  If you are aware of state-of-the-art algorithms that explicitly address uncertainty in merit, we would love to learn of the citations and perform comparisons for a final version.

---

### Decision · Program_Chairs · 2021-09-27

**Decision:**

Accept (Poster)

**Comment:**

After discussion, the reviewers were all in favor of accepting the paper. The discussion of related work should be expanded (in particular, KRW seems to ask for a very similar notion of fairness in rankings, although as you point out, the actual setting considered is distinct). Thanks for the strong submission!